# Sex determination gene *transformer* regulates the male-female difference in *Drosophila* fat storage via the adipokinetic hormone pathway

Lianna W Wat, Zahid S Chowdhury, Jason W Millington, Puja Biswas, Elizabeth J Rideout*

Department of Cellular and Physiological Sciences, The University of British Columbia, Vancouver, Canada

**Abstract** Sex differences in whole-body fat storage exist in many species. For example, *Drosophila* females store more fat than males. Yet, the mechanisms underlying this sex difference in fat storage remain incompletely understood. Here, we identify a key role for sex determination gene *transformer* (*tra*) in regulating the male-female difference in fat storage. Normally, a functional Tra protein is present only in females, where it promotes female sexual development. We show that loss of Tra in females reduced whole-body fat storage, whereas gain of Tra in males augmented fat storage. Tra's role in promoting fat storage was largely due to its function in neurons, specifically the Adipokinetic hormone (Akh)-producing cells (APCs). Our analysis of Akh pathway regulation revealed a male bias in APC activity and Akh pathway function, where this sex-biased regulation influenced the sex difference in fat storage by limiting triglyceride accumulation in males. Importantly, Tra loss in females increased Akh pathway activity, and genetically manipulating the Akh pathway rescued Tra-dependent effects on fat storage. This identifies sex-specific regulation of Akh as one mechanism underlying the male-female difference in whole-body triglyceride levels, and provides important insight into the conserved mechanisms underlying sexual dimorphism in whole-body fat storage.

*For correspondence:
elizabeth.rideout@ubc.ca

## Introduction

In animals, stored fat provides a rich source of energy to sustain basal metabolic processes to survive periods of nutrient scarcity, and to support reproduction (*Heier and Kühnlein, 2018*; *Heier et al., 2021*; *Walther and Farese, 2012*). The main form of stored fat is triglyceride, which is deposited within specialized organelles called lipid droplets (*Kühnlein, 2012*; *Murphy, 2001*; *Thiele and Spandl, 2008*). Lipid droplets are found in many cell types throughout the body, but the main organ responsible for triglyceride storage is the adipose tissue (*Murphy, 2001*). The amount of triglyceride in the adipose tissue is regulated by many factors; however, one important factor that influences an individual's whole-body fat level is whether the animal is female or male (*Karastergiou et al., 2012*; *Power and Schulkin, 2008*; *Sieber and Spradling, 2015*; *Wat et al., 2020*). Typically, females store more fat than males. In mammals, females store approximately 10% more body fat than males (*Jackson et al., 2002*; *Karastergiou et al., 2012*; *Womersley and Durnin, 1977*). Female insects, on the other hand, can store up to four times more fat than males of the same species (*Lease and Wolf, 2011*) and break down fat more slowly than males when nutrients are scarce (*Wat et al., 2020*). These male-female differences in fat metabolism play a key role in supporting successful reproduction in each sex: females with reduced fat storage often show lower fecundity (*Buszczak et al., 2002*; *Sieber and*

*Spradling, 2015*) whereas males with excess fat storage generally show decreased fertility (*Grönke et al., 2005*; *Wat et al., 2020*). Given that fat storage also influences diverse phenotypes such as immunity and lifespan (*DiAngelo and Birnbaum, 2009*; *Gáliková and Klepsatel, 2018*; *Johnson and Stolzing, 2019*; *Kamareddine et al., 2018*; *Liao et al., 2021*; *Roth et al., 2018*; *Suzawa et al., 2019*), the sex-specific regulation of fat storage has implications for several life-history traits. Yet, the genetic and physiological mechanisms that link biological sex with fat storage remain incompletely understood in many animals.

Clues into potential mechanisms underlying the sex difference in fat storage have emerged from studies on the regulation of triglyceride metabolism in *Drosophila*. While many pathways impact whole-body triglyceride levels (*Ballard et al., 2010*; *Bjedov et al., 2010*; *Broughton et al., 2005*; *Choi et al., 2015*; *DiAngelo and Birnbaum, 2009*; *Francis et al., 2010*; *Ghosh and O'Connor, 2014*; *Grönke et al., 2010*; *Heier and Kühnlein, 2018*; *Heier et al., 2021*; *Hentze et al., 2015*; *Kamareddine et al., 2018*; *Kang et al., 2017*; *Kubrak et al., 2020*; *Lee et al., 2019*; *Lehmann, 2018*; *Luong et al., 2006*; *Rajan and Perrimon, 2012*; *Roth et al., 2018*; *Scopelliti et al., 2019*; *Sieber and Spradling, 2015*; *Song et al., 2014*; *Song et al., 2017*; *Suzawa et al., 2019*; *Teleman et al., 2005*; *Texada et al., 2019*), the Adipokinetic hormone (Akh; FBgn0004552) pathway plays a central role in regulating whole-body fat storage and breakdown (*Heier and Kühnlein, 2018*; *Heier et al., 2021*; *Lehmann, 2018*). Akh is synthesized as a preprohormone in the Akh-producing cells (APCs), and is subsequently cleaved by proprotein convertases to produce active Akh (*Lee and Park, 2004*; *Noyes et al., 1995*; *Predel et al., 2004*; *Wegener et al., 2006*). When the APCs are activated by stimuli such as peptide hormones or neurons that make physical connections with the APCs (*Kubrak et al., 2020*; *Oh et al., 2019*; *Scopelliti et al., 2019*; *Zhao and Karpac, 2017*), Akh is released into the hemolymph (*Braco et al., 2012*).

Circulating Akh then interacts with a G-protein coupled receptor called the Akh receptor (AkhR, FBgn0025595), where Akh binding to AkhR on target tissues such as the fat body increases intracellular cyclic adenosine monophosphate (cAMP) levels. High levels of cAMP activate protein kinase A (PKA; FBgg0000242) (*Gäde and Auerswald, 2003*; *Park et al., 2002*; *Staubli et al., 2002*), which phosphorylates several downstream metabolic effectors to promote fat breakdown. For example, in insects, active PKA promotes fat breakdown via phosphorylation and activation of Lipid storage droplet-1 (Lsd-1; FBgn0039114) (*Arrese et al., 2008*; *Bickel et al., 2009*; *Gäde and Auerswald, 2003*; *Patel et al., 2005*). In mammals, fat breakdown is mediated by similar PKA-dependent phosphorylation of Perilipin 1, the mammalian homolog of Lsd-1, and by PKA-dependent phosphorylation and recruitment of lipases, such as Hormone-sensitive lipase (Hsl), to lipid droplets to promote fat mobilization (*Sztalryd and Brasaemle, 2017*). Given that these genes are highly conserved between mammals and flies (*Kühnlein, 2012*), similar PKA-dependent mechanisms likely explain triglyceride mobilization from lipid droplets. Thus, high levels of Akh pathway activity limit fat storage whereas low levels of Akh signaling promote fat storage. While Akh-mediated triglyceride breakdown plays a vital role in releasing stored energy during times of nutrient scarcity to promote survival (*Mochanová et al., 2018*), the Akh pathway limits fat storage even in contexts when nutrients are plentiful. Indeed, loss of *Akh* or *AkhR* augments fat storage in males under normal physiological conditions (*Bharucha et al., 2008*; *Gáliková et al., 2015*; *Grönke et al., 2007*), highlighting the critical role of this pathway in regulating whole-body triglyceride levels.

Additional clues into potential mechanisms underlying the sex difference in fat storage come from studies on metabolic genes. For example, flies carrying loss-of-function mutations in genes involved in triglyceride synthesis and storage, such as *midway* (*mdy*; FBgn0004797), *Lipin* (*Lpin*; FBgn0263593), *Lipid storage droplet-2* (*Lsd-2*; FBgn0030608), and *Seipin* (*Seipin*; FBgn0040336) show reduced whole-body triglyceride levels (*Buszczak et al., 2002*; *Grönke et al., 2003*; *Teixeira et al., 2003*; *Tian et al., 2011*; *Ugrankar et al., 2011*; *Wang et al., 2016*). Whole-body deficiency for genes that regulate triglyceride breakdown, on the other hand, generally have higher whole-body fat levels. This is best illustrated by elevated whole-body triglyceride levels found in flies lacking *brummer* (*bmm*; FBgn0036449) or *Hsl* (FBgn0034491), both of which encode lipases (*Bi et al., 2012*; *Grönke et al., 2005*). While these studies demonstrate the strength of *Drosophila* as a model in revealing conserved mechanisms that contribute to whole-body fat storage (*Recazens et al., 2021*; *Schreiber et al., 2019*; *Walther and Farese, 2012*), studies on *Drosophila* fat metabolism often use single- or mixed-sex groups of flies (*Bednářová et al., 2018*; *Gáliková et al., 2015*; *Grönke et al., 2007*; *Hughson et al.,*

*2021*; *Isabel et al., 2005*; *Lee and Park, 2004*; *Scopelliti et al., 2019*). As a result, less is known about how these metabolic genes and pathways contribute to the sex difference in fat storage.

Recent studies have begun to fill this knowledge gap by studying fat metabolism in both sexes. In one study, higher circulating levels of steroid hormone ecdysone in mated females were found to promote increased whole-body fat storage (*Sieber and Spradling, 2015*). Another study showed that elevated levels of *bmm* mRNA in male flies restricted triglyceride storage to limit whole-body fat storage (*Wat et al., 2020*). Yet, neither ecdysone signaling nor *bmm* fully explain known male-female differences in whole-body fat metabolism (*Sieber and Spradling, 2015*; *Wat et al., 2020*). This suggests additional metabolic genes and pathways must contribute to sex differences in fat storage and breakdown (*Wat et al., 2020*). Indeed, genome-wide association studies in *Drosophila* demonstrate sex-biased effects on fat storage for many genetic loci (*Nelson et al., 2016*; *Watanabe and Riddle, 2021*). As evidence of sex-specific mechanisms underlying whole-body fat storage continues to mount, several reports have also identified male-female differences in phenotypes linked with fat metabolism. For example, sex differences have been reported in energy physiology, metabolic rate, food intake, food preference, circadian rhythm, sleep, immune response, starvation resistance, and lifespan (*Andretic and Shaw, 2005*; *Austad and Fischer, 2016*; *Belmonte et al., 2019*; *Chandegra et al., 2017*; *Helfrich-Förster, 2000*; *Huber et al., 2004*; *Hudry et al., 2019*; *Millington et al., 2021*; *Park et al., 2018*; *Reddiex et al., 2013*; *Regan et al., 2016*; *Sieber and Spradling, 2015*; *Videlier et al., 2019*; *Wat et al., 2020*). More work is therefore needed to understand the genetic and physiological mechanisms underlying the male-female differences in fat metabolism, and to identify the impact of this sex-specific regulation on key life-history traits. Further, it will be critical to elucidate how these mechanisms are linked with upstream factors that determine sex.

In *Drosophila*, sexual development is determined by the number of X chromosomes (*Salz and Erickson, 2010*). In females, the presence of two X chromosomes triggers the production of a functional splicing factor called Sex-lethal (Sxl; FBgn0264270) (*Bell et al., 1988*; *Bridges, 1921*; *Cline, 1978*). Sxl's most well-known downstream target is *transformer* (*tra*; FBgn0003741), where Sxl-dependent splicing of *tra* pre-mRNA allows the production of a functional Tra protein (*Belote et al., 1989*; *Boggs et al., 1987*; *Inoue et al., 1990*; *Sosnowski et al., 1989*). In males, which have only one X chromosome, no functional Sxl or Tra proteins are made (*Cline and Meyer, 1996*; *Salz and Erickson, 2010*). Over several decades, a large body of evidence has accumulated showing that Sxl and Tra direct most aspects of female sexual identity, including effects on abdominal pigmentation, egg-laying, neural circuits, and behavior (*Anand et al., 2001*; *Baker et al., 2001*; *Billeter et al., 2006*; *Brown and King, 1961*; *Burtis and Baker, 1989*; *Camara et al., 2008*; *Christiansen et al., 2002*; *Cline, 1978*; *Cline and Meyer, 1996*; *Clough et al., 2014*; *Dauwalder, 2011*; *Demir and Dickson, 2005*; *Goodwin et al., 2000*; *Hall, 1994*; *Heinrichs et al., 1998*; *Hoshijima et al., 1991*; *Inoue et al., 1992*; *Ito et al., 1996*; *Nagoshi et al., 1988*; *Neville et al., 2014*; *Nojima et al., 2014*; *Pavlou et al., 2016*; *von Philipsborn et al., 2014*; *Pomatto et al., 2017*; *Rezával et al., 2014*; *Rezával et al., 2016*; *Rideout et al., 2007*; *Rideout et al., 2010*; *Ryner et al., 1996*; *Sturtevant, 1945*). More recently, studies have extended our knowledge of how Sxl and Tra regulate additional aspects of development and physiology such as body size and intestinal stem cell proliferation (*Ahmed et al., 2020*; *Hudry et al., 2016*; *Millington and Rideout, 2018*; *Millington et al., 2021*; *Regan et al., 2016*; *Rideout et al., 2015*; *Sawala and Gould, 2017*). Yet, the effects of sex determination genes on whole-body fat metabolism remain unknown, indicating a need for more knowledge of how factors that determine sexual identity influence this important aspect of physiology.

Here, we reveal a role for sex determination gene *tra* in regulating whole-body triglyceride storage. In females, Tra expression promotes a higher level of whole-body fat storage, whereas lack of a functional Tra protein in males leads to lower fat storage. Interestingly, neurons were the anatomical focus of *tra*'s effects on fat storage, where we show that ectopic Tra expression in male APCs was sufficient to augment whole-body triglyceride levels. Our analysis of Akh pathway regulation in both sexes revealed increased *Akh/AkhR* mRNA levels, APC activity, and Akh pathway activity in males. Our findings indicate that this overall male bias in the Akh pathway contributes to the sex difference in whole-body triglyceride levels by restricting fat storage in males. Importantly, we show that the presence of Tra influences Akh pathway activity, and that Akh lies genetically downstream of Tra in regulating whole-body fat storage. These results provide new insight into the mechanisms by which upstream

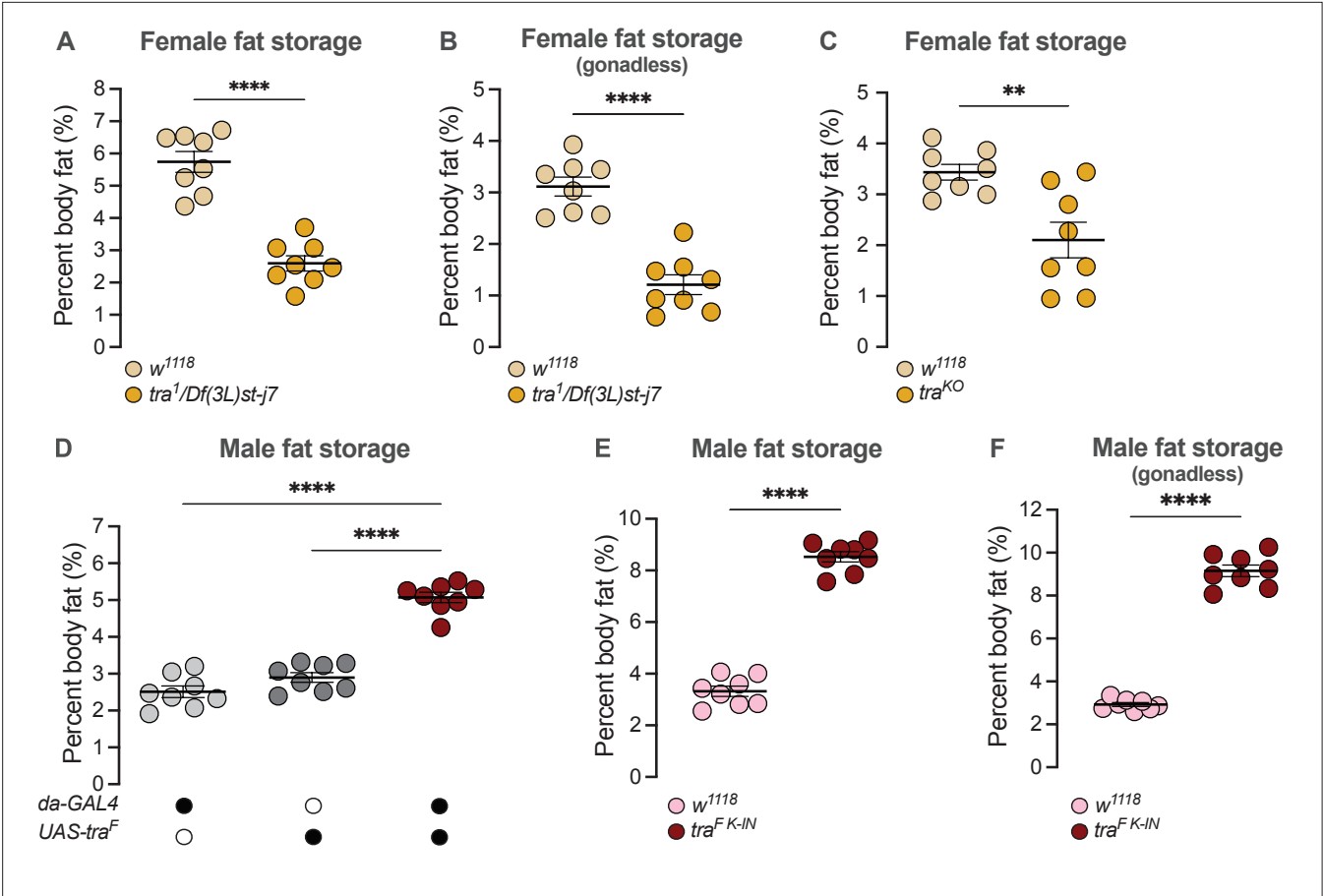

**Figure 1.** *transformer* regulates the sex difference in fat storage. (**A**) Whole-body triglyceride levels were significantly lower in *tra¹/Df(3 L)st-j7* females compared with *w¹¹¹⁸* controls (p<0.0001; Student's t-test). n=8 biological replicates. (**B**) Whole-body triglyceride levels were significantly lower in *tra¹/Df(3 L)st-j7* females with excised gonads compared with *w¹¹¹⁸* with excised ovaries (p<0.0001; Student's t-test). n=8 biological replicates. (**C**) Whole-body triglyceride levels were significantly lower in *tra^KO* females compared with *w¹¹¹⁸* controls (p=0.0037; Student's t-test). n=8 biological replicates. (**D**) Whole-body triglyceride levels were significantly higher in *da-GAL4>UAS-tra^F* males compared with *da-GAL4>+* and *+>UAS-tra^F* controls (p<0.0001 and p<0.0001, respectively; one-way ANOVA followed by Tukey's HSD). n=8 biological replicates. (**E**) Whole-body triglyceride levels were significantly higher in *tra^F K-IN* males compared with *w¹¹¹⁸* controls (p<0.0001, Student's t-test). n=8 biological replicates. (**F**) Whole-body triglyceride levels were significantly higher in *tra^F K-IN* males with excised gonads compared with *w¹¹¹⁸* controls lacking gonads (p<0.0001; Student's t-test). n=8 biological replicates. Black circles indicate the presence of a transgene and open circles indicate the lack of a transgene. ** indicates p<0.01, **** indicates p<0.0001; error bars represent SEM.

The online version of this article includes the following figure supplement(s) for figure 1:

**Figure supplement 1.** Elucidating *transformer*'s effect on sex differences in fat metabolism.

determinants of sexual identity, such as *tra*, influence the sex difference in fat storage. Further, we identify a previously unrecognized sex-biased role for Akh in regulating whole-body triglyceride levels.

## Results

### Sex determination gene *transformer* regulates the male-female difference in fat storage

Altered *Sxl* function in either sex causes significant lethality due to effects on the dosage compensation machinery (*Cline, 1978*; *Cline and Meyer, 1996*). We therefore asked whether the presence of Tra in females, which promotes female sexual development, contributes to the elevated whole-body triglyceride levels observed in females (*Sieber and Spradling, 2015*; *Wat et al., 2020*). In 5-day-old virgin females lacking *tra* function (*tra¹/Df(3 L)st-j7*), we found that whole-body triglyceride levels were significantly lower than in age-matched *w¹¹¹⁸* control females (*Figure 1A*). Because we

observed no significant difference in fat storage between *tra¹/Df(3 L)st-j7* mutant males and *w¹¹¹⁸* control males (*Figure 1—figure supplement 1A*), the sex difference in whole-body triglyceride storage was reduced. While previous studies show the ovaries store a small amount of triglyceride (*Sieber and Spradling, 2015*; *Wat et al., 2020*), Tra's effect on whole-body triglyceride storage was not explained by the absence of ovaries in females lacking Tra function: whole-body fat storage was still significantly lower in *tra¹/Df(3 L)st-j7* mutant females with excised gonads compared with *w¹¹¹⁸* control females with excised ovaries (*Figure 1B*). Given that we reproduced this finding in females carrying a distinct combination of *tra* mutant alleles (*Figure 1C*; *Hudry et al., 2016*), our findings suggest Tra regulates the sex difference in whole-body triglyceride levels by promoting fat storage in females.

We next asked whether Tra function also contributes to the reduced fat breakdown phenotype post-starvation in females (*Wat et al., 2020*). To quantify fat breakdown, we measured whole-body triglyceride levels at 0 hr , and 24 hr after food withdrawal, and calculated the percent change in whole-body triglyceride levels between time points. While female flies normally have reduced fat breakdown post-starvation compared with males (*Wat et al., 2020*), the magnitude of fat breakdown post-starvation was not significantly different between *tra¹/Df(3 L)st-j7* mutants and sex-matched *w¹¹¹⁸* controls (genotype:time interactions p=0.6298 [females], p=0.3853 [males]; *Supplementary file 1*; *Figure 1—figure supplement 1B*). Tra function is therefore required to promote elevated fat storage in females, but does not regulate fat breakdown post-starvation.

Given that males normally lack a functional Tra protein (*Belote et al., 1989*; *Boggs et al., 1987*; *Inoue et al., 1990*; *Sosnowski et al., 1989*), we next asked whether the absence of Tra in males explains their reduced whole-body triglyceride levels and rapid triglyceride breakdown post-starvation (*Wat et al., 2020*). To test this, we ubiquitously overexpressed Tra using *daughterless* (*da*)-*GAL4*, an established way to feminize male flies (*Ferveur et al., 1995*; *Rideout et al., 2015*), and examined whole-body fat metabolism. In 5-day-old *da-GAL4>UAS-traᶠ* males, whole-body triglyceride levels were significantly higher than in age-matched *da-GAL4>+* or *+>UAS-traᶠ* control males (*Figure 1D*). No increase in whole-body fat storage was observed in age-matched *da-GAL4>UAS-traᶠ* females compared with *da-GAL4>+* or *+>UAS-traᶠ* control females (*Figure 1—figure supplement 1C*); therefore, the sex difference in fat storage was reduced. Because high levels of Tra overexpression affected viability in one study (*Siera and Cline, 2008*), we also measured fat storage in males carrying an allele of *tra* that directs the production of physiological Tra levels (*traᶠ K-IN* allele) (*Hudry et al., 2019*). As in *da-GAL4>UAS-traᶠ* males, whole-body triglyceride levels were significantly higher in *traᶠ K-IN* males compared with *w¹¹¹⁸* control males (*Figure 1E*). While these data indicate that gain of a functional Tra protein in males promotes whole-body fat storage, we note that the magnitude of the increase in fat storage was higher in *traᶠ K-IN* males. The reason for this discrepancy between *tra*-expressing males is not clear, therefore, future studies will need to compare *tra* expression levels and tissue distribution between *da-GAL4>UAS-traᶠ* males and *traᶠ K-IN* males.

Importantly, the presence of rudimentary ovaries in *traᶠ K-IN* males did not explain their increased fat storage, as whole-body fat storage was still higher in *traᶠ K-IN* males lacking gonads compared with gonadless control males (*Figure 1F*). The elevated fat storage in *traᶠ K-IN* males also cannot be attributed to ecdysone production by the rudimentary ovaries, as no ecdysone target genes were upregulated (*Figure 1—figure supplement 1D*; *Sieber and Spradling, 2015*); however, future studies will need to address why these *traᶠ K-IN* males showed significant ecdysone target gene downregulation. Taken together, these data indicate that lack of Tra function contributes to the reduced whole-body triglyceride levels normally observed in males. In males, this role for Tra may also extend to the regulation of fat breakdown, as triglyceride mobilization post-starvation was significantly reduced in *da-GAL4>UAS-traᶠ* males compared with *da-GAL4>+* or *+>UAS-traᶠ* controls during a 24-hr starvation period (genotype:time p<0.0001 [males]; *Supplementary file 1*; *Figure 1—figure supplement 1E*), a finding we reproduced in *traᶠ K-IN* males (*Figure 1—figure supplement 1F*). While this effect of Tra on fat breakdown in males does not perfectly align with our data from *tra* mutant females, we note a trend toward increased fat breakdown in *tra* mutant females that was not statistically significant (*Figure 1—figure supplement 1B*). Taken together, these data support a clear role for Tra in regulating the sex difference in fat storage, and suggest that a role for Tra in regulating fat breakdown cannot be ruled out.

## *transformer* function in neurons regulates the sex difference in fat storage

Tra function is required in many cell types, tissues, and organs to promote female sexual development (*Anand et al., 2001*; *Baker et al., 2001*; *Billeter et al., 2006*; *Brown and King, 1961*; *Burtis and Baker, 1989*; *Camara et al., 2008*; *Christiansen et al., 2002*; *Clough et al., 2014*; *Dauwalder, 2011*; *Demir and Dickson, 2005*; *Goodwin et al., 2000*; *Hall, 1994*; *Heinrichs et al., 1998*; *Hoshijima et al., 1991*; *Inoue et al., 1992*; *Ito et al., 1996*; *Nagoshi et al., 1988*; *Neville et al., 2014*; *Nojima et al., 2014*; *Pavlou et al., 2016*; *von Philipsborn et al., 2014*; *Pomatto et al., 2017*; *Rezával et al., 2014*; *Rezával et al., 2016*; *Rideout et al., 2007*; *Rideout et al., 2010*; *Ryner et al., 1996*; *Sturtevant, 1945*). To determine the cell types and tissues in which Tra function is required to influence fat metabolism, we overexpressed Tra using a panel of GAL4 lines that drive expression in subsets of cells and/or tissues. To rapidly assess potential effects on fat metabolism, we measured starvation resistance, an established readout for changes to fat storage and breakdown (*Beller et al., 2010*; *Bi et al., 2012*; *Choi et al., 2015*; *Grönke et al., 2003*; *Grönke et al., 2005*; *Grönke et al., 2007*; *Gutierrez et al., 2007*).

Normally, adult females have elevated starvation resistance compared with age-matched males due to higher fat storage and reduced fat breakdown (*Wat et al., 2020*). Indeed, loss of *tra* reduced starvation resistance in females (*Figure 2A*) whereas gain of Tra function enhanced starvation resistance in males (*Figure 2B*), in line with their effects on fat metabolism (*Figure 1A and D*). From our survey of different GAL4 lines (*Figure 2—figure supplement 1A-F*; *Figure 2—figure supplement 2A-D*), we found that neurons were the cell type in which gain of Tra most strongly extended male starvation resistance (*Figure 2C*). Specifically, starvation resistance in males with Tra overexpression in neurons (*elav-GAL4>UAS-tra^F*) was significantly extended compared with *elav-GAL4>+* and *+>UAS-tra^F* controls (*Figure 2C*), with no effect in females (*Figure 2—figure supplement 3A*). Because the increase in starvation resistance upon neuron-specific Tra expression was similar in magnitude to the increase in survival observed upon global Tra expression (*Figure 2B and C*), this finding suggests a key role for neuronal Tra in regulating starvation resistance.

To determine whether increased starvation resistance in *elav-GAL4>UAS-tra^F* males was due to altered fat metabolism, we measured whole-body triglyceride levels in males and females with neuronal Tra overexpression. We found that *elav-GAL4>UAS-tra^F* males (*Figure 2D*), but not females (*Figure 2—figure supplement 3B*), showed a significant increase in whole-body fat storage compared with sex-matched *elav-GAL4>+* and *+>UAS-tra^F* controls. This suggests that the male-specific increase in starvation resistance (*Figure 2C*) was due to increased fat storage in *elav-GAL4>UAS-tra^F* males, which we confirm by showing that the rate of fat breakdown in *elav-GAL4>UAS-tra^F* males and females was not significantly different from sex-matched *elav-GAL4>+* and *+>UAS-tra^F* controls (*Figure 2—figure supplement 3C*) (genotype:time interaction p=0.2789 [males], p=0.7058 [females]; *Supplementary file 1*). Neurons are therefore one cell type in which Tra function influences the sex difference in whole-body triglyceride storage.

To identify specific neurons that mediate Tra's effects on starvation resistance and whole-body fat storage, we overexpressed Tra in neurons known to affect fat metabolism and measured starvation resistance (*Figure 2—figure supplement 4A-E*; *Al-Anzi and Zinn, 2018*; *Al-Anzi et al., 2009*; *Chung et al., 2017*; *Li et al., 2016*; *May et al., 2020*; *Min et al., 2016*; *Mosher et al., 2015*; *Zhan et al., 2016*). One group of neurons that significantly augmented starvation resistance upon Tra expression was the APCs (*Figure 2E*), a group of neuroendocrine cells in the corpora cardiaca that produce Akh and other peptide hormones such as Limostatin (Lst; FBgn0034140) (*Alfa et al., 2015*; *Lee and Park, 2004*). Although we note that Tra expression in additional neurons and in glia affected starvation resistance (*Figure 2—figure supplement 2D*; *Figure 2—figure supplement 4D*), suggesting the regulation of fat metabolism by Tra function in neurons is complex, the central role of the APCs in regulating fat metabolism prompted a more detailed investigation into Tra's function in these neurons. Flies with APC-specific Tra expression (*Akh-GAL4>UAS-tra^F*) had significantly increased starvation resistance compared with sex-matched *Akh-GAL4>+* and *+>UAS-tra^F* controls (*Figure 2E*; *Figure 2—figure supplement 5A*). To determine whether the starvation resistance phenotype indicated altered fat storage, we compared whole-body triglyceride levels in *Akh-GAL4>UAS-tra^F* males and females with sex-matched *Akh-GAL4>+* and *+>UAS-tra^F* controls. There was a significant increase in whole-body fat storage in males (*Figure 2F*) but not females (*Figure 2—figure supplement 5B*) with APC-specific

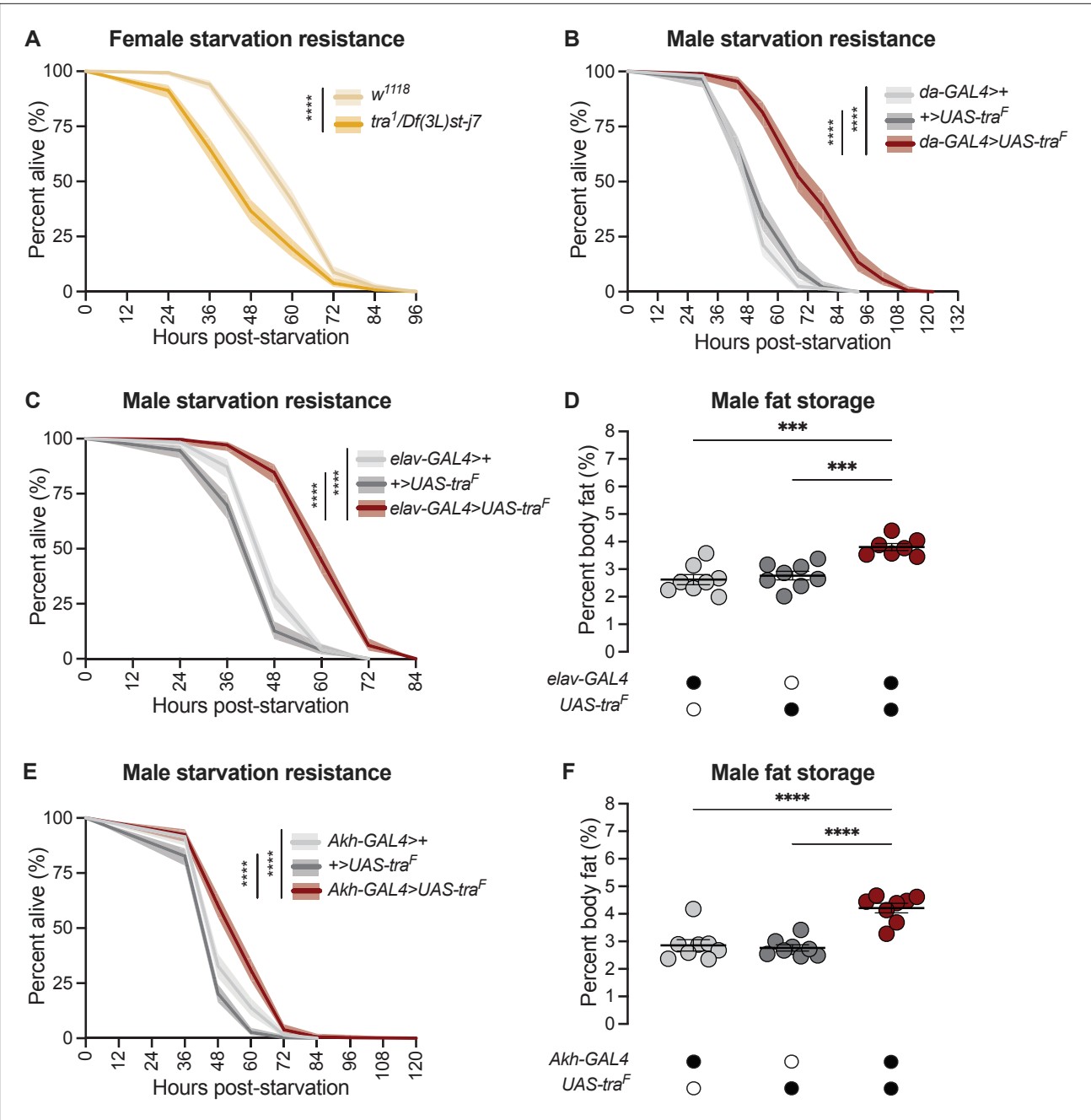

**Figure 2.** *transformer* function in Akh-producing cells contributes to the sex difference in fat storage. (**A**) Starvation resistance was significantly reduced in *tra$^1$/Df(3 L)st-j7* females compared with *w$^{1118}$* controls (p<2×10$^{-16}$; log-rank test, Bonferroni's correction for multiple comparisons). n=344–502 animals. (**B**) Starvation resistance was significantly enhanced in *da-GAL4>UAS-tra$^F$* males compared with *da-GAL4>+* and *+>UAS-tra$^F$* controls (p<2×10$^{-16}$ and p<2×10$^{-16}$, respectively; log-rank test, Bonferroni's correction for multiple comparisons). n=198–201 animals. (**C**) Starvation resistance was significantly enhanced in *elav-GAL4>UAS-tra$^F$* males compared with *elav-GAL4>+* and *+>UAS-tra$^F$* controls (p<2×10$^{-16}$ and p<2×10$^{-16}$, respectively; log-rank test, Bonferroni's correction for multiple comparisons). n=248–279 animals. (**D**) Whole-body triglyceride levels were significantly higher in *elav-GAL4>UAS-tra$^F$* males compared with *elav-GAL4>+* and *+>UAS-tra$^F$* controls (p=0.0001 and p=0.0006, respectively; one-way ANOVA followed by Tukey's HSD). n=7–8 biological replicates. (**E**) Starvation resistance was significantly enhanced in *Akh-GAL4>UAS-tra$^F$* males compared with *Akh-GAL4>+* and *+>UAS-tra$^F$* controls (p=3.1×10$^{-9}$ and p<2×10$^{-16}$, respectively; log-rank test, Bonferroni's correction for multiple comparisons). n=280–364 animals. (**F**) Whole-body triglyceride levels were significantly higher in *Akh-GAL4>UAS-tra$^F$* males compared to *Akh-GAL4>+* and *+>UAS-tra$^F$* control males (p<0.0001 and p<0.0001, respectively; one-way ANOVA followed by Tukey's HSD). n=8 biological replicates. Black circles indicate the presence of a transgene and open circles indicate the lack of a transgene. *** indicates p<0.001, **** indicates p<0.0001; shaded areas represent the 95% confidence interval; error bars represent SEM.

*Figure 2 continued on next page*

*Figure 2 continued*

The online version of this article includes the following figure supplement(s) for figure 2:

**Figure supplement 1.** Effect of *transformer* gain in multiple cell types and tissues on starvation resistance.

**Figure supplement 2.** Effect of *transformer* gain in additional cell types and tissues on starvation resistance.

**Figure supplement 3.** Gain of *transformer* function in neurons does not affect fat breakdown.

**Figure supplement 4.** Effect of *transformer* gain in multiple neuronal subsets on starvation resistance.

**Figure supplement 5.** Gain of *transformer* function in Akh-producing cells does not affect fat breakdown.

Tra expression. This indicates Tra function in the APCs promotes fat storage, revealing a previously unrecognized role for the APCs in regulating the sex difference in fat storage. Indeed, fat breakdown was unaffected in *Akh-GAL4>UAS-tra^F* males and females compared with sex-matched *Akh-GAL4>+* and *+>UAS-tra^F* controls (*Figure 2—figure supplement 5C*) (genotype:time interaction p=0.1201 [males] and p=0.0596 [females]; *Supplementary file 1*).

## Sex-specific regulation of adipokinetic hormone leads to a male bias in pathway activity

Given that the sexual identity of the APCs impacts whole-body fat storage, we compared the regulation of *Akh*, APC activity, and Akh signaling between adult males and females. We first examined *Akh* and *AkhR* mRNA levels in both sexes using quantitative real-time polymerase chain reaction (qPCR). We found that mRNA levels of both *Akh* and *AkhR* were significantly higher in 5-day-old *w^1118* males than in females (*Figure 3A and B*). This male bias in *Akh* mRNA levels did not reflect an increased APC number in males, as we found no sex difference in the number of APCs (*Figure 3C*). Because Akh release from the APCs is regulated by APC activity (*Kubrak et al., 2020*; *Oh et al., 2019*), we next measured APC activity in males and females by driving APC-specific expression of calcium-responsive chimeric transcription factor *LexA-VP16-NFAT* (*Akh-GAL4>UAS-LexA-VP16-NFAT* [called *UAS-CaLexA*]) (*Masuyama et al., 2012*). Sustained APC activity triggers nuclear import of LexA-VP16-NFAT, where it drives expression of a GFP reporter downstream of a LexA-responsive element (*Masuyama et al., 2012*). Monitoring GFP levels in the APCs therefore provides a straightforward way to monitor APC activity.

In 5-day-old *Akh-GAL4>UAS-CaLexA* males, GFP levels were significantly higher than in age- and genotype-matched females (*Figure 3D–H*). Because *GAL4* mRNA levels were not significantly different between males and females carrying the *Akh-GAL4* transgene (*Figure 3—figure supplement 1A*), and the number of APCs did not differ between the sexes (*Figure 3C*), these findings indicate that the APCs are more active in males than in females. To determine whether the male bias in *Akh/AkhR* mRNA levels and APC activity affected Akh pathway activity, we next compared levels of phosphorylated Inositol-requiring enzyme-1 (Ire1; FBgn0261984) between males and females. Because levels of phosphorylated Ire1 (p-Ire1) are higher in *Drosophila* cells stimulated with Akh peptide, regulation that was dependent on the presence of *AkhR*, high p-Ire1 levels indicate increased Akh pathway activity (*Song et al., 2017*). We found that the ratio of p-Ire1 to loading control actin was higher in 5-day-old *w^1118* males compared with age- and genotype-matched females in three out of four biological replicates (*Figure 3I–K*; *Figure 3—figure supplement 1B*), a finding that aligns with the sex difference in *Akh/AkhR* mRNA levels and APC activity. Taken together, our data suggest a previously unrecognized male bias in the Akh pathway.

## The adipokinetic hormone pathway contributes to the sex difference in fat storage

Given that high Akh pathway activity limits fat storage via an established intracellular signaling cascade that culminates in lipase recruitment and fat mobilization (*Baumbach et al., 2014*; *Grönke et al., 2007*; *Lee and Park, 2004*; *Mochanová et al., 2018*), we wanted to determine whether the male bias in Akh pathway activity influences the sex difference in fat metabolism by restricting fat storage in males. We therefore used a published approach to ablate the APCs (*Akh-GAL4>UAS-reaper* (*rpr*)) (*Lee and Park, 2004*; *White et al., 1996*), and measured whole-body triglyceride levels in each sex. Because the sexual identity of the APCs affects fat storage and not fat breakdown (*Figure 2F*;

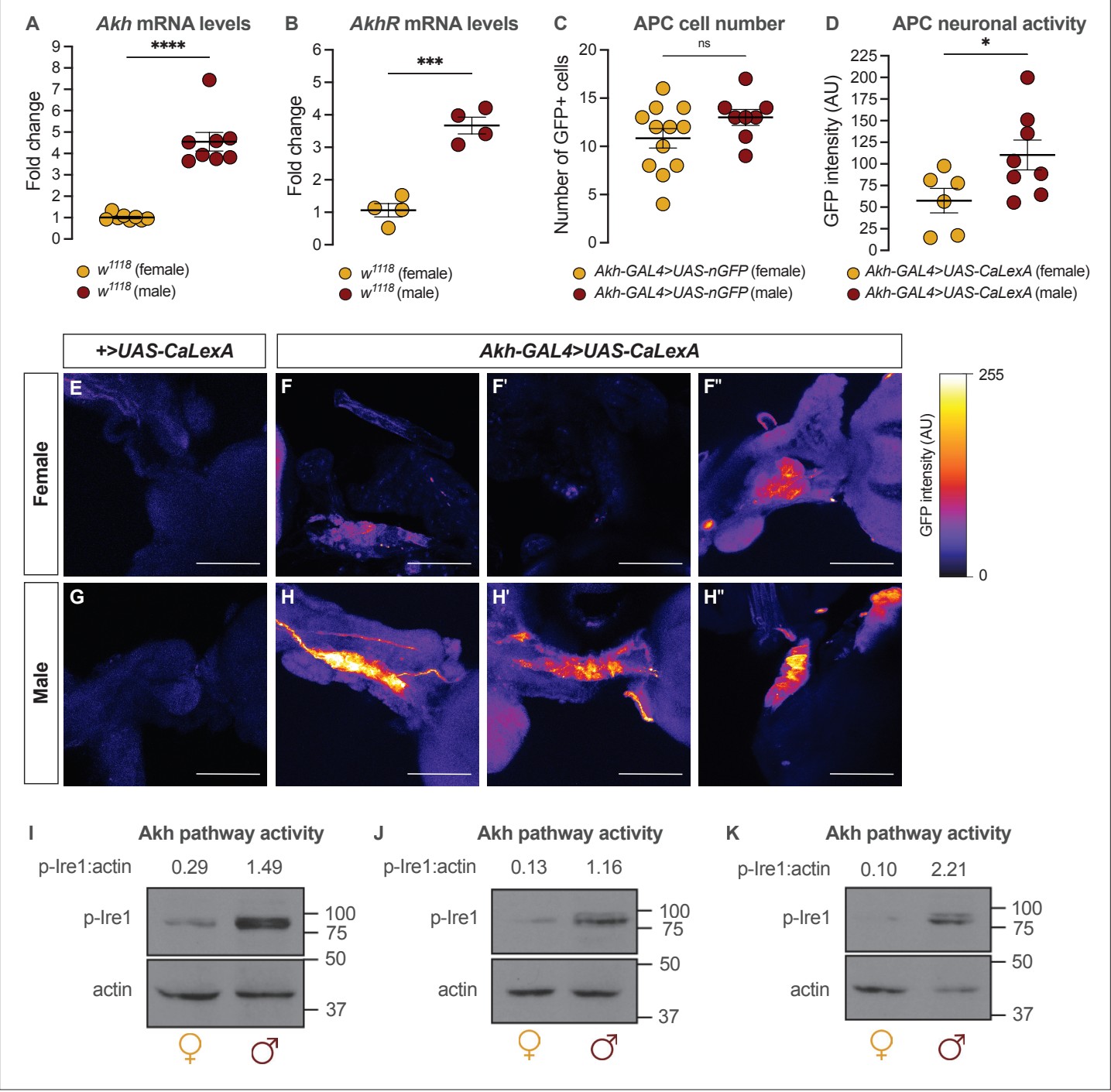

**Figure 3.** Sex-specific regulation of Akh and the Akh signaling pathway. (**A**) *Akh* mRNA levels were significantly higher in $w^{1118}$ males compared with genotype-matched females (p<0.0001, Student's t-test). n=8 biological replicates. (**B**) *AkhR* mRNA levels were significantly higher in $w^{1118}$ males than in females (p=0.0002, Student's t-test). n=4 biological replicates. (**C**) Expression of *UAS-nGFP* in Akh-producing cells (APCs) (*Akh-GAL4>UAS* -nGFP) revealed no significant difference in APC cell number between males and females (p=0.1417; Student's t-test). n=8–12 animals. (**D**) GFP intensity produced as a readout of calcium activity in the APCs (*Akh-GAL4>LexAop-CD8-GFP;UAS-LexA-VP16-NFAT (UAS-CaLexA)*) was significantly higher in males compared with females (p=0.0438; Student's t-test). n=6–8 biological replicates. (**E–H**) Maximum Z-projections of representative images showing GFP produced as a readout for APC calcium activity from both *Akh-GAL4>UAS-CaLexA* males and females. Scale bars=50 μm, n=6–8 biological replicates. (**I–K**) Whole-body p-Ire1 levels were higher in $w^{1118}$ males compared with $w^{1118}$ females in three biological replicates. * indicates p<0.05, *** indicates p<0.001, **** indicates p<0.0001, ns indicates not significant; error bars represent SEM. Original images for (**C**) are found in *Figure 3—source data 1*. Original images for (**D–H**) are found in *Figure 3—source data 2*. Original images for (**I–K**) are found in *Figure 3—source data 3*.

*Figure 3 continued on next page*

*Figure 3 continued*

The online version of this article includes the following figure supplement(s) for figure 3:

**Source data 1.** Images used to quantify number of Akh-producing cells.

**Source data 2.** Images used to quantify neuronal activity of Akh-producing cells.

**Source data 3.** Original blots for p-Ire1 and actin in males and females.

**Figure supplement 1.** *Akh-GAL4* drives equivalent *GAL4* mRNA levels in both sexes.

**Figure supplement 1—source data 1.** Original blots for p-Ire1 and actin in male versus female.

---

*Figure 2—figure supplement 5C*), we focused our analysis on triglyceride storage rather than mobilization. Triglyceride levels were significantly higher in 5-day-old *Akh-GAL4>UAS -rpr* males than in *Akh-GAL4>+* and *+>UAS-rpr* control males (*Figure 4A*). In contrast, triglyceride levels in 5-day-old *Akh-GAL4>UAS -rpr* females were not significantly different from *Akh-GAL4>+* and *+>UAS-rpr* control females (*Figure 4—figure supplement 1A*). This suggests that the male bias in Akh pathway activity normally contributes to the sex difference in fat storage by limiting triglyceride accumulation in males via the established intracellular signaling cascade known to regulate lipid droplet breakdown (*Arrese et al., 2008*; *Heier and Kühnlein, 2018*; *Heier et al., 2021*; *Kühnlein, 2012*; *Patel et al., 2005*). Importantly, we reproduced the male-biased effects on fat storage in flies carrying loss-of-function *Akh* and *AkhR* alleles (*Akh[A]* and *AkhR[1]*, respectively) (*Figure 4B and C*; *Figure 4—figure supplement 1B, C*), and show that APC-specific knockdown of *Lst* had no effect on fat storage in either sex (*Figure 4—figure supplement 1D, E*). These findings support a model in which it is Akh production by the APCs that plays a role in regulating the male-female difference in fat storage. While the mechanisms underlying the regulation of intracellular fat breakdown by Akh in the fat body have been well-documented (*Bharucha et al., 2008*; *Gáliková et al., 2015*; *Grönke et al., 2007*; *Heier and Kühnlein, 2018*; *Heier et al., 2021*; *Kubrak et al., 2020*; *Lee and Park, 2004*; *Lehmann, 2018*; *Scopelliti et al., 2019*; *Zhao and Karpac, 2017*), our findings reveal a new role for Akh in regulating the sex difference in fat storage. Notably, this Akh-mediated regulation of the male-female difference in fat storage operates in a parallel pathway to the previously described sex-specific role of triglyceride lipase *bmm* (*Figure 4—figure supplement 2A, B*; *Wat et al., 2020*).

Beyond the APC ablation or complete loss of Akh, we next wanted to test whether the sex-specific Akh regulation we uncovered contributes to the male-female difference in fat storage. To this end, we used a genetic approach to manipulate *Akh* mRNA levels or APC activity, and measured whole-body fat storage in both sexes. To determine whether the male bias in *Akh* mRNA levels contributes to the sex difference in fat storage, we measured whole-body triglyceride levels in flies with APC-specific expression of *Akh-RNAi* (*Akh-GAL4>UAS-Akh-RNAi*). Importantly, this manipulation effectively reduced *Akh* mRNA levels in both sexes (*Figure 4—figure supplement 3A,B*). In males, whole-body triglyceride levels were significantly higher in *Akh-GAL4>UAS-Akh-RNAi* flies compared with *Akh-GAL4>+* and *+>UAS-Akh-RNAi* control flies (*Figure 4D*). *Akh-GAL4>UAS-Akh-RNAi* female flies, in contrast, showed no significant change in whole-body fat storage compared with *Akh-GAL4>+* and *+>UAS-Akh-RNAi* control females (*Figure 4—figure supplement 3C*). This indicates a strongly male-biased effect on fat storage due to reduced *Akh* mRNA levels, suggesting that the sex difference in *Akh* mRNA levels contributes to the male-female difference in whole-body fat storage.

To determine whether the male bias in APC activity also influences the sex difference in fat storage, we silenced the APCs by APC-specific overexpression of an inwardly rectifying potassium channel Kir2.1 (*Baines et al., 2001*) and measured whole-body triglyceride levels. Whole-body fat storage in *Akh-GAL4>UAS-Kir2.1* adult males was significantly higher compared with *Akh-GAL4>+* and *+>UAS-Kir2.1* control males (*Figure 4E*). In females, while we observed significantly elevated whole-body fat storage in *Akh-GAL4>UAS-Kir2.1* adults compared with *Akh-GAL4>+* and *+>UAS-Kir2.1* controls (*Figure 4—figure supplement 3D*), the magnitude of this increase was larger in males (sex:genotype interaction p=0.0455; *Supplementary file 1*). Taken together, these data suggest that the male bias in APC activity contributes to the sex difference in fat storage by limiting triglyceride accumulation in males. Indeed, augmenting APC activity in females using a bacterial voltage-gated sodium channel (*UAS-NaChBac*) significantly reduced fat storage in females (*Figure 4F*; *Figure 4—figure supplement 3E*). While Akh affects food-related behaviors in some contexts (*Choi et al., 2015*; *Hentze et al., 2015*; *Huang et al., 2020*), we observed no significant effects of altered APC activity on feeding behavior

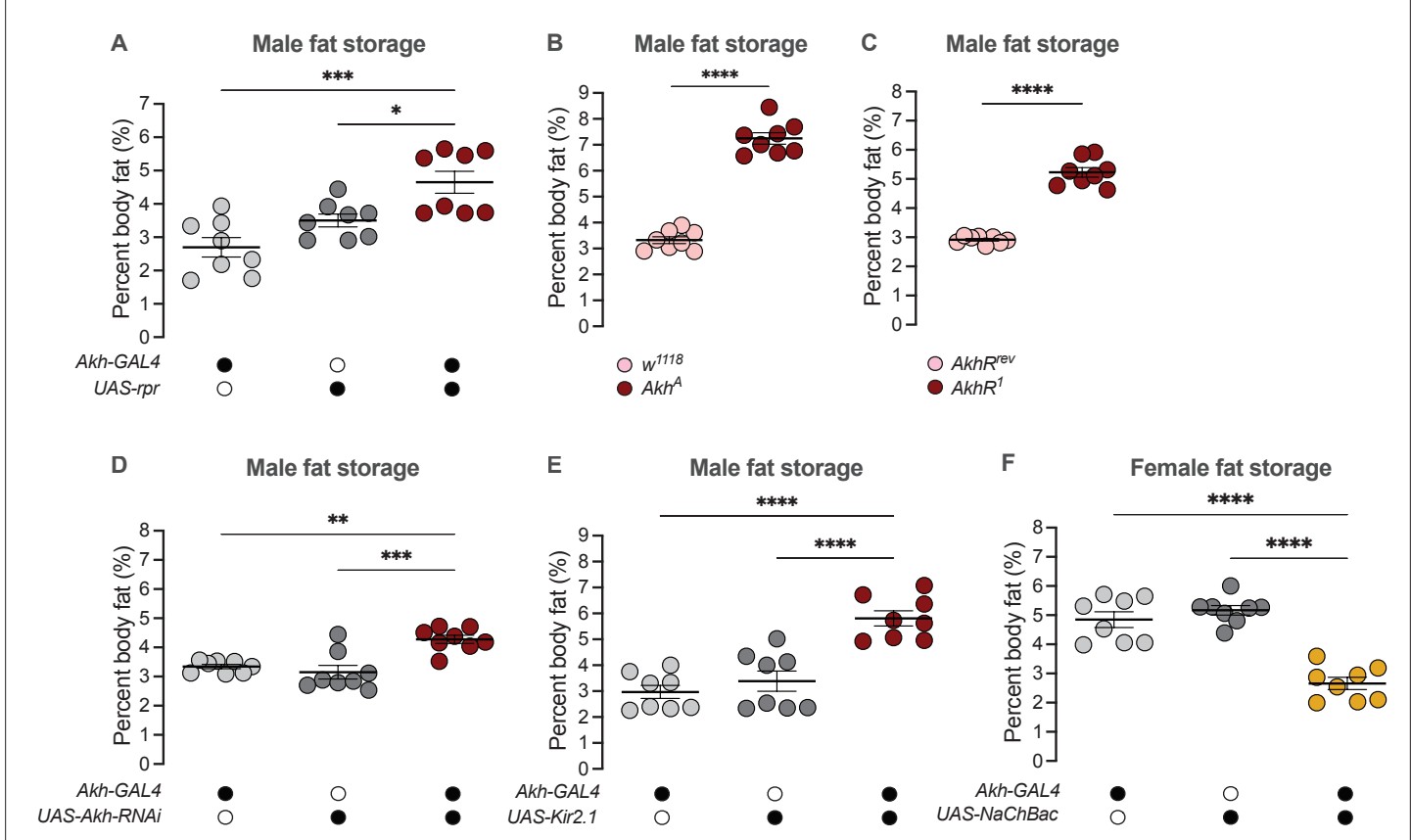

**Figure 4.** Sex-specific regulation of Akh and APC activity influence the sex difference in fat storage. (**A**) Whole-body triglyceride levels were significantly higher in *Akh-GAL4>UAS-reaper (rpr)* males compared with *Akh-GAL4>+* and *+>UAS-rpr* controls (p=0.0002 and p=0.0215, respectively; one-way ANOVA followed by Tukey's HSD). n=8 biological replicates. (**B**) Whole-body triglyceride levels were significantly higher in *AkhA* males compared with *w1118* controls (p<0.0001; one-way ANOVA followed by Tukey's HSD). n=8 biological replicates. (**C**) Whole-body triglyceride levels were significantly higher in *AkhR1* males compared with *AkhRrev* controls (p<0.0001; one-way ANOVA followed by Tukey's HSD). n=8 biological replicates. (**D**) Whole-body triglyceride levels were significantly higher in *Akh-GAL4>UAS-Akh-RNAi* males compared with *Akh-GAL4>+* and *+>UAS-Akh-RNAi* controls (p=0.0015 and p=0.0002, respectively; one-way ANOVA followed by Tukey's HSD). n=8 biological replicates. (**E**) Whole-body triglyceride levels were significantly higher in *Akh-GAL4>UAS-Kir2.1* males compared with *Akh-GAL4>+* and *+>UAS-Kir2.1* controls (p<0.0001 and p<0.0001, respectively; one-way ANOVA followed by Tukey's HSD). n=8 biological replicates. (**F**) Whole-body triglyceride levels were significantly lower in *Akh-GAL4>UAS-NaChBac* females compared with *Akh-GAL4>+* and *+>UAS-NaChBac* controls (p<0.0001 and p<0.0001, respectively; one-way ANOVA followed by Tukey's HSD). n=8 biological replicates. Due to independent experiments with a shared GAL4 control, *Akh-GAL4>+* males are shared between (**E**) and *Figure 4—figure supplement 3E*. *Akh-GAL4>+* females are shared between (**F**) and *Figure 4—figure supplement 3D*. Black circles indicate the presence of a transgene and open circles indicate the lack of a transgene; * indicates p<0.05, ** indicates p<0.01, *** indicates p<0.001, **** indicates p<0.0001; error bars represent SEM.

The online version of this article includes the following figure supplement(s) for figure 4:

**Figure supplement 1.** APC-derived Limostatin does not regulate the sex difference in fat storage.

**Figure supplement 2.** Akh and *brummer* operate in parallel pathways to regulate the sex difference in fat storage.

**Figure supplement 3.** RNAi-mediated Akh knockdown effectively reduced Akh transcripts in both sexes.

**Figure supplement 4.** Activity of the Akh-producing cells does not regulate food consumption in either sex.

in either sex (*Figure 4—figure supplement 4A-D*). This suggests that the male-biased effect of APC manipulation on fat storage cannot be fully explained by effects on food intake. Thus, in addition to the contribution of elevated *Akh* mRNA levels in males to the sex difference in fat storage, we also identify a role for the male bias in APC activity in the sex-specific regulation of whole-body triglyceride levels.

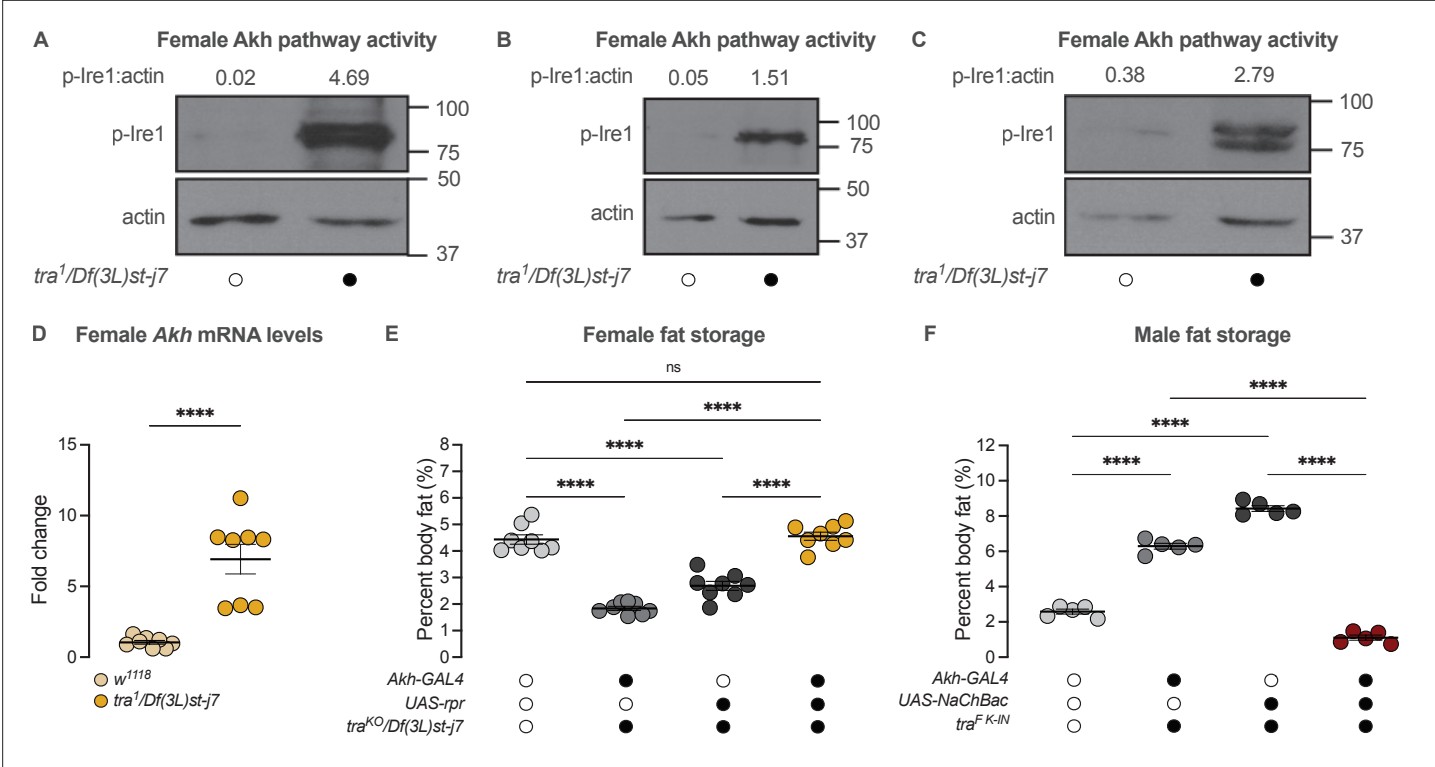

**Figure 5.** *transformer* regulates the sex difference in fat storage via the Akh signalling pathway. (**A–C**) Whole-body p-Ire1 levels were higher in *tra[1]/Df(3 L)st-j7* females compared with *w[1118]* controls in three biological replicates. (**D**) Whole-body *Akh* mRNA levels were significantly higher in *tra[1]/Df(3 L)st-j7* females compared with *w[1118]* controls (p<0.0001; Student's t-test). n=8 biological replicates. (**E**) Whole-body triglyceride levels were significantly lower in *tra[KO]/Df(3 L)st-j7* females carrying either *Akh-GAL4>+* or *+>UAS-reaper* (*rpr*) transgenes compared with *w[1118]* controls carrying a functional Tra protein (p<0.0001 and p<0.0001, respectively; one-way ANOVA followed by Tukey's HSD). Whole-body triglyceride levels were not significantly different between *tra[KO]/Df(3 L)st-j7* females lacking APCs (*Akh-GAL4>UAS-rpr*) and *w[1118]* controls (p=0.9384; one-way ANOVA followed by Tukey's HSD). n=8 biological replicates. (**F**) Whole-body triglyceride levels were significantly higher in *tra[F K-IN]* males carrying either *Akh-GAL4>+* or *+>UAS-NaChBac* transgenes compared with *w[1118]* control males lacking Tra function (p<0.0001 and p<0.0001, respectively; one-way ANOVA followed by Tukey's HSD). Whole-body triglyceride levels in *tra[F K-IN]* males with APC activation (*Akh-GAL4>UAS-NaChBac*) were significantly lower than *tra[F K-IN]* males carrying either the *Akh-GAL4>+* or *+>UAS-NaChBac* transgenes alone (p<0.0001 and p<0.0001, respectively; one-way ANOVA followed by Tukey's HSD). n=5 biological replicates. Black circles indicate the presence of a transgene or mutant allele and open circles indicate the lack of a transgene or mutant allele. **** indicates p<0.0001, ns indicates not significant; error bars represent SEM. Original images for (**A–C**) are found in *Figure 5—source data 1*.

The online version of this article includes the following figure supplement(s) for figure 5:

**Source data 1.** Original blots for p-Ire1 and actin in females with whole-body loss of *transformer*.

**Figure supplement 1.** Whole-body p-Ire1 levels in *transformer* mutant flies.

**Figure supplement 1—source data 1.** Original blots for p-Ire1 and actin in females with whole body loss of *transformer*.

## *transformer* regulates the sex difference in fat storage via the adipokinetic hormone pathway

Given that Tra function and the Akh pathway both contribute to the male-female difference in fat storage, we asked whether the presence of Tra affects the sex bias in Akh pathway activity. In 5-day-old *tra[1]/Df(3 L)st-j7* females, levels of p-Ire1 were higher than in *w[1118]* control females in three out of four biological replicates (*Figure 5A–C*; *Figure 5—figure supplement 1A*). This suggests the presence of Tra in females normally represses Akh pathway activity. Indeed, loss of Tra significantly increased *Akh* mRNA levels in females (*Figure 5D*). Given Tra's effects on Akh pathway activity, we next tested whether the change in Akh pathway function was significant for Tra's effects on whole-body triglyceride levels. We predicted that if increased Akh pathway activity caused the lower fat storage in *tra* mutant females, genetic manipulations that reduce Akh pathway activity should block this reduction in whole-body triglyceride levels. While all female genotypes lacking *tra* function had reduced fat storage compared with control females (*Figure 5E*), APC ablation in *tra* mutant females

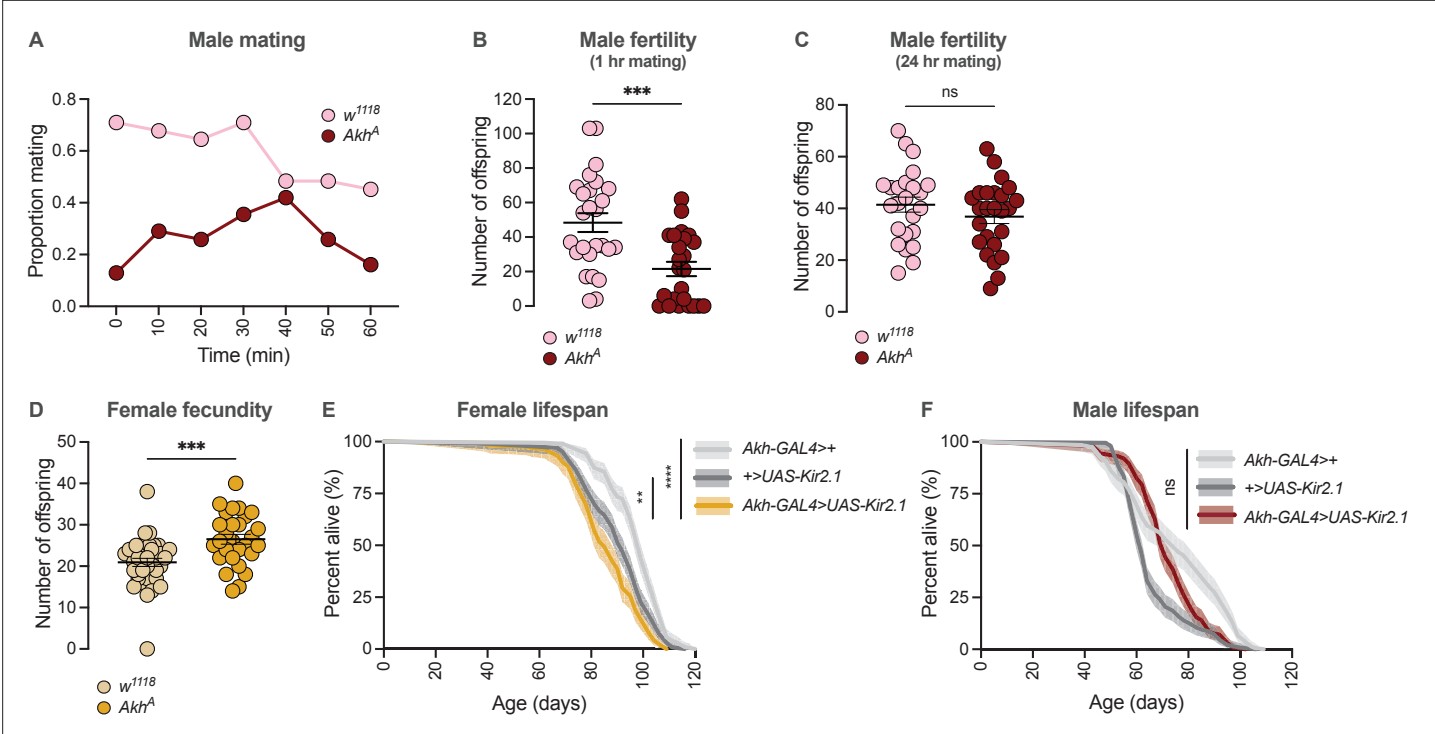

**Figure 6.** Sex-specific regulation of Akh signalling pathway promotes reproductive success in each sex. (**A**) At all observation points, a lower proportion of *Akh^A* males were successfully copulating with a wildtype *Canton-S* (*CS*) female compared with *w^1118* controls. n=31 males. (**B**) The number of pupae produced from a 60-min mating period was significantly lower in *Akh^A* males compared with *w^1118* controls (p=0.0003; Student's t-test). n=24–26 males. (**C**) The number of pupae produced from a 24-hr mating period was not significantly different between *Akh^A* males and *w^1118* control males (p=0.2501; Student's t-test). n=24–25 males. (**D**) The number of pupae produced from a 24-hr mating period was significantly higher in *Akh^A* females compared with *w^1118* controls (p=0.0006; Student's t-test). n=28–36 females. (**E**) Lifespan was significantly shorter in *Akh-GAL4>UAS-Kir2.1* females compared with *Akh-GAL4>+* and *+>UAS-Kir2.1* controls (p<2×10^{-16} and p=0.0015, respectively; log-rank test, Bonferroni's correction for multiple comparisons). n=160–198 females. (**F**) Lifespan of *Akh-GAL4>UAS-Kir2.1* males was intermediate between *Akh-GAL4>+* and *+>UAS-Kir2.1* controls, indicating no overall effect of inhibiting APC neuronal activity on male lifespan (p=0.00013 and p=7.0×10^{-6}, respectively; log-rank test, Bonferroni's correction for multiple comparisons). n=196–200 males. ** indicates p<0.01, *** indicates p<0.001, **** indicates p<0.0001, ns indicates not significant; error bars represent SEM; shaded areas represent the 95% confidence interval.

rescued this decrease in whole-body triglyceride levels (*Figure 5E*). Indeed, fat storage in *tra* mutant females lacking APCs was not significantly different from *w^1118* control females (p=0.9384; *Supplementary file 1*; *Figure 5E*), indicating that the increased Akh pathway activity we observed in *tra* mutant females was one reason for their reduced fat storage. Given that APC activation in males expressing physiological levels of Tra similarly rescued the Tra-induced increase in whole-body triglyceride levels (*Figure 5F*), these findings suggest that the sex-specific regulation of Akh pathway activity represents one way *tra* influences the male-female difference in fat storage.

## Loss of adipokinetic hormone has opposite effects on reproductive success in each sex and mediates a fecundity-lifespan tradeoff in females

Our results suggest that adult females show lower Akh pathway activity and higher fat storage, whereas males maintain a higher level of Akh activity and lower fat storage. Because the correct regulation of fat storage in each sex influences reproduction (*Buszczak et al., 2002*; *Grönke et al., 2005*; *Sieber and Spradling, 2015*; *Wat et al., 2020*), we tested how complete loss of this critical regulator of the sex difference in fat storage impacted offspring production in each sex. In *Akh^A* mutant males, we found that the proportion of males copulating with a *Canton-S (CS)* virgin female was lower than in control *w^1118* males at each 10 min interval during a 60-min observation period (*Figure 6A*). When we counted viable offspring from these copulation events, we found that *Akh^A* mutant males had significantly fewer overall progeny than *w^1118* control males (*Figure 6B*). These results suggest that Akh

function normally promotes reproductive success in males; however, it is important to note that Akh function is not absolutely required for male fertility, as a prolonged 24 hr period of contact between *Akh*[A] mutant males and *CS* females allowed the production of normal progeny numbers (*Figure 6C*).

In contrast to males, Akh loss-of-function mutations in females increased fecundity (*Figure 6D*). Specifically, *Akh*[A] mutant females produced a significantly higher number of offspring compared with *w*[1118] controls (*Figure 6D*). Thus, in females, a low level of Akh pathway activity promotes fecundity. Given that a change in one life-history trait such as reproduction often affects traits such as longevity (*Chapman et al., 1995*; *Flatt, 2011*; *Fowler and Partridge, 1989*; *Hansen et al., 2013*), we also measured lifespan in females with reduced Akh pathway function. We found that lifespan was significantly shorter in *Akh-GAL4>UAS-Kir2.1* females compared with *Akh-GAL4>+* and *+>UAS-Kir2.1* control females (*Figure 6E*). In contrast, male lifespan was not significantly different between *Akh-GAL4>UAS-Kir2.1* flies and *Akh-GAL4>+* and *+>UAS-Kir2.1* controls (*Figure 6F*). Our findings are in agreement with a previous study that demonstrated a female-specific lifespan reduction in response to whole-body loss of *Akh* (*Bednářová et al., 2018*). This suggests that while low Akh activity in females promotes fertility, this benefit comes at the cost of a shorter lifespan, a possibility that will

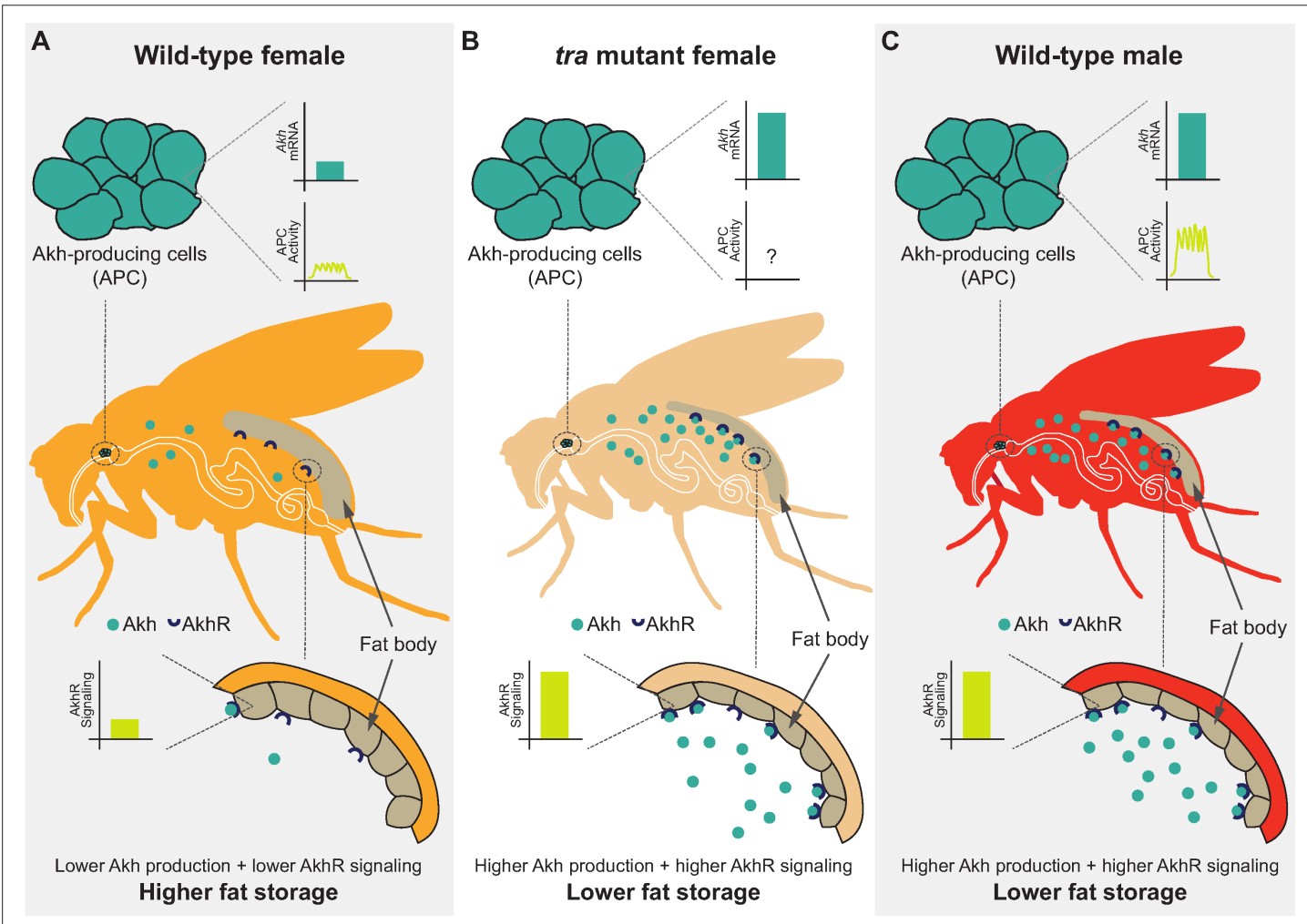

**Figure 7.** Sex-specific regulation of the Akh pathway by *tra* contributes to the sex difference in fat storage. (**A**) In wild-type females, *Akh* mRNA transcripts and APC activity are lower compared with wild-type males, leading to lower AkhR signaling. Given that AkhR signaling stimulates fat breakdown, lower AkhR signaling in females contributes to higher female fat storage. (**B**) In females lacking functional *tra*, *Akh* mRNA transcripts are higher compared with wild-type females, leading to higher AkhR signaling. Higher AkhR signaling in *tra* mutant females contributes to lower *tra* mutant female fat storage. (**C**) In wild-type males, *Akh* mRNA transcripts and APC activity are higher compared with wild-type females, leading to higher AkhR signaling. Higher AkhR signaling in males contributes to lower male fat storage.

be explored in future studies using additional strains to genetically augment, or inhibit, Akh pathway activity (e.g., APC activation, *Akh* mutants).

## Discussion

In this study, we used the fruit fly *Drosophila melanogaster* to improve the knowledge of the mechanisms underlying the male-female difference in whole-body triglyceride levels. We show that the presence of a functional Tra protein in females, which directs many aspects of female sexual development, promotes whole-body fat storage. Tra's ability to promote fat storage arises largely due to its function in neurons, where we identified the APCs as one neuronal population in which Tra function influences whole-body triglyceride levels. Our examination of *Akh/AkhR* mRNA levels and APC activity revealed several differences between the sexes, where these differences lead to higher Akh pathway activity in males than in females (*Figure 7A and C*). Genetic manipulation of APCs and Akh pathway activity suggest a model in which the sex bias in Akh pathway activity contributes to the male-female difference in fat storage by limiting whole-body triglyceride storage in males (*Figure 7C*). Importantly, we show that Tra function influences Akh pathway activity, and that Akh acts genetically downstream of Tra in regulating whole-body triglyceride levels (*Figure 7B*). This reveals a previously unrecognized genetic and physiological mechanism that contributes to the sex difference in fat storage.

One key finding from our study was the identification of sex determination gene *tra* as an upstream regulator of the male-female difference in fat storage. In females, a functional Tra protein promotes fat storage, whereas lack of Tra in males leads to reduced fat storage. While an extensive body of literature has demonstrated important roles for *tra* in regulating neural circuits, behavior, abdominal pigmentation, and gonad development (*Anand et al., 2001*; *Baker et al., 2001*; *Billeter et al., 2006*; *Brown and King, 1961*; *Burtis and Baker, 1989*; *Camara et al., 2008*; *Christiansen et al., 2002*; *Clough et al., 2014*; *Dauwalder, 2011*; *Demir and Dickson, 2005*; *Goodwin et al., 2000*; *Hall, 1994*; *Heinrichs et al., 1998*; *Hoshijima et al., 1991*; *Inoue et al., 1992*; *Ito et al., 1996*; *Nagoshi et al., 1988*; *Neville et al., 2014*; *Nojima et al., 2014*; *Pavlou et al., 2016*; *von Philipsborn et al., 2014*; *Pomatto et al., 2017*; *Rezával et al., 2014*; *Rezával et al., 2016*; *Rideout et al., 2007*; *Rideout et al., 2010*; *Ryner et al., 1996*; *Sturtevant, 1945*), uncovering a role for *tra* in regulating fat storage significantly extends our understanding of how sex differences in metabolism arise. Given that sex differences exist in other aspects of metabolism (e.g., oxygen consumption) (*Wat et al., 2020*), this new insight suggests that more work will be needed to determine whether *tra* contributes to sexual dimorphism in additional metabolic traits. Indeed, one study showed that *tra* influences the sex difference in adaptation to hydrogen peroxide stress (*Pomatto et al., 2017*). Beyond metabolism, Tra also regulates multiple aspects of development and physiology such as intestinal stem cell proliferation (*Ahmed et al., 2020*; *Hudry et al., 2016*; *Millington and Rideout, 2018*), carbohydrate metabolism (*Hudry et al., 2019*), body size (*Mathews et al., 2017*; *Rideout et al., 2015*), phenotypic plasticity (*Millington et al., 2021*), and lifespan responses to dietary restriction (*Regan et al., 2016*). Because some, but not all, of these studies identify a cell type in which Tra function influences these diverse phenotypes, future studies will need to determine which cell types and tissues require Tra expression to establish a female metabolic and physiological state. Indeed, recent single-cell analyses reveal widespread gene expression differences in shared cell types between the sexes (*Li et al., 2021*).

Identifying neurons as the anatomical focus of Tra's effects on fat storage was another key finding from our study. While many sexually dimorphic neural circuits related to behavior and reproduction have been identified (*Anand et al., 2001*; *Auer and Benton, 2016*; *Baker et al., 2001*; *Billeter et al., 2006*; *Clyne and Miesenböck, 2008*; *Dauwalder, 2011*; *Demir and Dickson, 2005*; *Evans and Cline, 2007*; *Goodwin et al., 2000*; *Hall, 1994*; *Inoue et al., 1992*; *Ito et al., 1996*; *Kimura et al., 2019*; *Kvitsiani and Dickson, 2006*; *Neville et al., 2014*; *Nojima et al., 2014*; *Pavlou et al., 2016*; *von Philipsborn et al., 2014*; *Rezával et al., 2014*; *Rezával et al., 2016*; *Rideout et al., 2007*; *Rideout et al., 2010*; *Ryner et al., 1996*; *Sato et al., 2019*; *Shirangi et al., 2016*; *Wang et al., 2020*), less is known about sex differences in neurons that regulate physiology and metabolism. Indeed, while many studies have identified neurons that regulate fat metabolism (*Al-Anzi and Zinn, 2018*; *Al-Anzi et al., 2009*; *Chung et al., 2017*; *Li et al., 2016*; *May et al., 2020*; *Min et al., 2016*; *Mosher et al., 2015*; *Zhan et al., 2016*), these studies were conducted in single- or mixed-sex populations. Because male-female differences in neuron number (*Billeter et al., 2006*; *Castellanos et al., 2013*; *Demir and Dickson, 2005*; *Garner et al., 2017*; *Lee and Hall, 2001*; *Rideout et al., 2007*; *Rideout et al.,*

2010; *Robinett et al., 2010*; *Taylor and Truman, 1992*), morphology (*Cachero et al., 2010*; *Kimura et al., 2019*), activity (*Guo et al., 2016*), and connectivity (*Cachero et al., 2010*; *Nojima et al., 2021*) have all been described across the brain and ventral nerve cord (*Mellert et al., 2010*; *Mellert et al., 2016*), a detailed analysis of neuronal populations that influence metabolism will be needed in both sexes to understand how neurons contribute to the sex-specific regulation of metabolism and physiology. Indeed, while our identification of a role for APC sexual identity in regulating the male-female difference in fat storage represents a significant step forward in understanding how sex differences in neurons influence metabolic traits, more knowledge is needed of how Tra regulates sexual dimorphism in this critical neuronal subset. For example, while we show that females normally have lower *Akh* mRNA levels and APC activity, it remains unclear how the presence of Tra regulates these distinct traits. Tra may regulate *Akh* mRNA levels via known target genes *fruitless* (*fru*; FBgn0004652) and *doublesex* (*dsx*; FBgn0000504) (*Burtis and Baker, 1989*; *Heinrichs et al., 1998*; *Hoshijima et al., 1991*; *Inoue et al., 1992*; *Nagoshi et al., 1988*; *Ryner et al., 1996*), or alternatively through a *fru*- and *dsx*-independent pathway (*Hudry et al., 2016*; *Rideout et al., 2015*). To influence the sex difference in APC activity and Akh release, Tra may regulate factors such as ATP-sensitive potassium ($K_{ATP}$) channels and 5′ adenosine monophosphate-activated protein kinase (AMPK)-dependent signaling, both of which are known to modulate APC activity (*Braco et al., 2012*; *Kim and Rulifson, 2004*). Future studies will therefore need to investigate Tra-dependent changes to $K_{ATP}$ channel expression and function in APCs, and characterize Tra's effects on ATP levels and AMPK signaling within APCs.

Additional ways to learn more about the sex-specific regulation of fat storage by the APCs will include examining how sexual identity influences physical connections between the APCs and other neurons, and monitoring APC responses to circulating hormones. For example, there are physical connections between corazonin- and neuropeptide F (NPF; FBgn0027109)-positive (CN) neurons and APCs in adult male flies (*Oh et al., 2019*), and between the APCs and a bursicon-α-responsive subset of DLgr2 neurons in females (*Scopelliti et al., 2019*). These connections inhibit APC activity: CN neurons inhibit APC activity in response to high hemolymph sugar levels (*Oh et al., 2019*), whereas binding of bursicon-α to DLgr2 neurons inhibits APC activity in nutrient-rich conditions (*Scopelliti et al., 2019*). Future studies will therefore need to determine whether these physical connections exist in both sexes. Further, it will be important to identify male-female differences in circulating factors that regulate the APCs. While gut-derived Allatostatin C (AstC; FBgn0032336) was recently shown to bind its receptor on the APCs to trigger Akh release, loss of AstC affects fat metabolism and starvation resistance only in females (*Kubrak et al., 2020*). This suggests sex differences in AstC-dependent regulation of fat metabolism may exist.

Given that gut-derived NPF binds to its receptor on the APCs to inhibit Akh release (*Yoshinari et al., 2021*),that skeletal muscle-derived unpaired 2 (upd2; FBgn0030904) regulates hemolymph Akh levels (*Zhao and Karpac, 2017*), and that circulating peptides such as Allatostatin A (AstA; FBgn0015591), *Drosophila* insulin-like peptides (Dilps), and activin ligands influence Akh pathway activity (*Ahmad et al., 2020*; *Hentze et al., 2015*; *Post et al., 2019*; *Song et al., 2017*), it is clear that a systematic survey of circulating factors that modulate Akh production, release, and Akh pathway activity in each sex will be needed to fully understand the sex-specific regulation of fat storage. Another important point to address in future studies will be confirming results from previous studies that the fat body is the main anatomical focus of Akh-dependent regulation of fat storage (*Bharucha et al., 2008*; *Grönke et al., 2007*). Given that the sex-biased effects of triglyceride lipase *bmm* arise from a male-female difference in the cell type-specific requirements for *bmm* function (*Wat et al., 2020*), it will be important to determine which cell types mediate Akh's effects on fat storage in each sex. This line of enquiry will also clarify the underlying processes that support increased fat storage in females. At present, it remains unclear whether the higher whole-body fat storage in females is caused by lower fat breakdown (*Wat et al., 2020*), increased lipogenesis, or both. Given that Akh pathway activity plays a role in regulating both lipolysis and lipogenesis in *Drosophila* and other insects (*Grönke et al., 2007*; *Lee and Goldsworthy, 1995*; *Lorenz, 2003*), it will be important to identify the cellular mechanism underlying Akh's effects on the sex difference in fat storage.

Beyond fat metabolism, it will be important to extend our understanding of how sex-specific Akh regulation affects additional Akh-regulated phenotypes. Given that we and others show Akh affects fertility and fecundity (*Liao et al., 2021*), future studies will need to determine whether these phenotypes are due to Akh-dependent regulation of fat metabolism, or due to direct effects of Akh on

gonads. Similarly, while Akh has been linked with the regulation of lifespan (*Bednářová et al., 2018*; *Liao et al., 2021*), carbohydrate metabolism (*Kim and Rulifson, 2004*; *Lee and Park, 2004*), starvation resistance (*Isabel et al., 2005*; *Kubrak et al., 2020*; *Mochanová et al., 2018*), locomotion (*Isabel et al., 2005*; *Lee and Park, 2004*), immune responses (*Adamo et al., 2008*), cardiac function (*Isabel et al., 2005*; *Noyes et al., 1995*), and oxidative stress responses (*Gáliková et al., 2015*), most studies were performed in mixed- or single-sex populations. Additional work is therefore needed to determine how changes to Akh pathway function affect physiology, carbohydrate levels, development, and life history in each sex. Importantly, the lessons we learn may also extend to other species. Akh signalling is highly conserved across invertebrates (*Gäde and Auerswald, 2003*; *Lorenz and Gäde, 2009*; *Staubli et al., 2002*), and is functionally similar to the mammalian β-adrenergic and glucagon systems (*Grönke et al., 2007*; *Lee and Park, 2004*; *Staubli et al., 2002*). Because sex-specific regulation of both glucagon and the β-adrenergic systems have been described in mammalian models and in humans (*Al-Gburi et al., 2017*; *Bell et al., 2001*; *Bilginoglu et al., 2007*; *Brooks et al., 2015*; *Claustre et al., 1980*; *Dart et al., 2002*; *Davis et al., 2000*; *Drake et al., 1998*; *Freedman et al., 1987*; *Hinojosa-Laborde et al., 1999*; *Hoeker et al., 2014*; *Hogarth et al., 2007*; *Lafontan et al., 1997*; *Luzier et al., 1998*; *McIntosh et al., 2011*; *Ng et al., 1993*), detailed studies on sex-specific Akh regulation and function in flies may provide vital clues into the mechanisms underlying male-female differences in physiology and metabolism in other animals.

## Materials and methods

### Fly husbandry

Fly stocks were maintained at 25°C in a 12:12 light:dark cycle. All larvae were reared at a density of 50 larvae per 10 ml of fly media (recipe in *Supplementary file 4*). Males and females were separated either as early pupae by gonad size, or late pupae by the presence of sex combs. Sex-transformed males and females were distinguished by the presence (males) or absence (females) of B$^S$Y. Single-sex groups of 20 pupae were transferred to damp filter paper within a food vial until eclosion. Unless otherwise stated, all experiments used 5- to 7-day-old flies.

### Fly strains

We obtained the following strains from the Bloomington *Drosophila* Stock Center: *Canton-S* (#64349), *w$^{1118}$* (#3605), *UAS-nGFP* (#4775), *UAS-Akh-RNAi* (#27031), *UAS-tra$^F$* (#4590), *tra$^1$* (#675), *Df(3 L)st-j7* (#5416), *UAS-NaChBac* (#9468), *UAS-Kir2.1* (#6595), *UAS-reaper* (#5823), and *UAS-CaLexA* (#66542). We obtained *Akh$^A$*, *AkhR$^{rev}$*, *AkhR$^1$*, *bmm$^1$*, and *AkhR$^1$;bmm$^1$* as kind gifts from Dr. Ronald Kühnlein (*Gáliková et al., 2015*; *Grönke et al., 2005*; *Grönke et al., 2007*), *tra$^{KO}$* and *tra$^{F\ K-IN}$* as kind gifts from Dr. Irene Miguel-Aliaga (*Hudry et al., 2016*; *Hudry et al., 2019*), and *Mex-GAL4* as a kind gift from Dr. Claire Thomas (*Phillips and Thomas, 2006*). The authors acknowledge critical resources and information provided by Flybase (*Thurmond et al., 2019*). The following GAL4 lines were used for tissue-specific expression: *da-GAL4* (ubiquitous), *cg-GAL4* (fat body), *r4-GAL4* (fat body), *Lsp2-GAL4* (fat body), *Myo1A-GAL4* (enterocytes), *Mex-GAL4* (enterocytes), *dMef2-GAL4* (skeletal muscle), *repo-GAL4* (glia), *elav-GAL4* (neurons), *c587-GAL4* (somatic cells of the gonad), *tj-GAL4* (somatic cells of the gonad), *nos-GAL4* (germ cells of the gonad), *dimmed-GAL4* (peptidergic neurons), *TH-GAL4* (dopaminergic neurons), *Tdc2-GAL4* (octopaminergic neurons), *VT030559-GAL4* (mushroom body neurons), *dilp2-GAL4* (insulin-producing cells), and *Akh-GAL4* (APCs). All transgenic stocks were backcrossed into a *w$^{1118}$* background for a minimum of five generations.

### Adult weight

To measure adult weight, groups of 10 flies were weighed in 1.5 ml microcentrifuge tubes on an analytical balance (Mettler-Toledo, ME104).

### RNA extraction, cDNA synthesis, and qPCR

One biological replicate consisted of five flies homogenized in 200 µl of TRIzol. RNA was extracted following the manufacturer's instructions, as previously described (*Wat et al., 2020*). cDNA was synthesized from RNA using the Quantitect Reverse Transcription Kit (Qiagen, 205311). qPCR was

used to quantify relative mRNA transcript levels as previously described (*Wat et al., 2020*). See *Supplementary file 3* for a full list of primers.

## Whole-body triglyceride measurements

One biological replicate consisted of five flies homogenized in 200 µl of 0.1% Tween (AMresco, 0777-1 L) in 1× phosphate-buffered saline (PBS) using 50 µl of glass beads (Sigma-Aldrich, 11079110) agitated at 8 m/s for 5 s (OMNI International Bead Ruptor 24). Assay was performed according to established protocols (*Tennessen et al., 2014*) as previously described (*Wat et al., 2020*).

## Gonad excision

Five-day-old adult flies were individually anesthetized with $CO_2$. The gonads or ovaries were removed from the distal end of the abdomen in cold 1× PBS and the carcass was snap-frozen in a 1.5 ml microcentrifuge tube on dry ice.

## Western blotting

One biological replicate consisted of 10 flies homogenized in extraction buffer (females=200 µl, males=125 µl) containing 20 mM Hepes (pH 7.8), 450 mM NaCl, 25% glycerol, 50 mM NaF, 0.2 mM EDTA, 0.5% Triton X-100, 1 mM PMSF, 1 mM DTT, 1× cOmplete Protease Inhibitor Cocktail (Roche), and 1× PhosSTOP (Roche) using 50 µl of glass beads (Sigma-Aldrich, 11079110) agitated at 8 m/s for 5 s (OMNI International Bead Ruptor 24). Samples were incubated on ice for 5 min before cellular debris was pelleted by centrifugation at 10,000 rpm for 5 min at 4 °C and supernatant was removed (Thermo Fisher Scientific, Heraeus Pico 21 centrifuge). Centrifugation was repeated two times more to remove fat from the samples. Protein concentration of each sample was determined by a Bradford Assay (Bio-Rad, 550-0205); 20 µg of protein per sample was loaded onto a 12% SDS-PAGE gel. Immunoblotting was performed as previously described (*Millington et al., 2021*). Primary antibodies used were rabbit anti-p-Ire1 (1:1000; Abcam #48187) and mouse anti-actin (1:200; Santa Cruz #sc-8432). Secondary antibodies used were goat anti-rabbit (1:5000; Invitrogen #65-6120) and horse anti-mouse (1:2000; Cell Signaling Technology #7076).

## APC measurements

To isolate the APCs, individual flies were anesthetized on ice, and the brain and foregut were removed in cold 1× PBS. Samples were fixed in 4% paraformaldehyde for 30 min, followed by two 30 min washes in cold 1× PBS. Samples were incubated with Hoechst (Sigma-Aldrich, 33342) at a concentration of 1:500 for 5 min and mounted in SlowFade Diamond Antifade Mountant (Thermo Fisher Scientific, S36967). Images were captured using a Leica TCS SP5 Confocal Microscope and processed using Fiji (ImageJ; *Schindelin et al., 2012*). To visualize APC neuronal activity (*Akh-GAL4>UAS-CaLexA*), the mean GFP intensity of one APC cluster was quantified by measuring average pixel intensity within the region of interest using Fiji (ImageJ; *Schindelin et al., 2012*). To determine APC cell number (*Akh-GAL4>UAS -nGFP*), GFP punctae were counted manually using Fiji (ImageJ; *Schindelin et al., 2012*). One biological replicate consists of one cluster of APCs, where only one APC cluster was measured per individual.

## Capillary feeder assay

One biological replicate consisted of 10 flies placed into a specialized 15 ml conical vial with access to two capillary tubes. Capillary tubes were filled with fly food media containing 5% sucrose, 5% yeast extract, 0.3% propionic acid, and 0.15% nipagin. Approximately 0.5 µl of mineral oil was added to the top of each capillary tube to prevent evaporation. All vials were placed into fitted holes in the lid of a large plastic container. A shallow layer of water was poured into the base of the container to maintain high humidity throughout the experiment. The meniscus of the fly food media was marked before the start of the experiment and again after 24 hr. The distance between the marks is used to quantify the volume of fly food media that was consumed (1 mm=0.15 µl). The volume of fly food consumed was normalized to the weight of individual flies (protocol adapted from *Stafford et al., 2012*).

## Male fertility

One singly housed male was placed with a group of three virgin *Canton-S (CS)* females and allowed to interact for 60 min. At 10 min intervals during the 60-min observation period, we recorded whether

a copulating male-female pair was present in the vial. After the 60-min observation period, the male was removed from the vial and the females were allowed to lay eggs for 72 hr (flies were transferred to new food every 24 hr). After 72 hr, the females were removed and progeny were allowed to develop. After 10 days, we counted the number of pupae in each vial. For the 24 hr mating assay, one singly housed male was allowed to interact with three virgin *CS* females for 24 hr before the male was removed and females were allowed to lay eggs for 72 hr as described above.

### Female fecundity

One virgin female was placed with a group of three virgin *CS* males for 24 hr. The females were then transferred onto fresh food every 24 hr for 3 days and the number of pupae was counted as described above.

### Starvation assays

Five-day-old flies were transferred to vials containing 2 ml of starvation media ( 0.75 agar in 1× PBS). To measure fat breakdown post-starvation, biological replicates consisting of five flies each were collected in 1.5 ml microcentrifuge tubes and snap-frozen on dry ice at 0 hr and 24 hr post-starvation. The percent change in fat storage between time points was calculated to determine fat breakdown over time. For starvation resistance, the number of deaths was recorded every 12 hr until no living flies remained in the vial.

### Lifespan

Flies were transferred to new vials with 2 ml of fresh food every 2–3 days until no living flies remained in the vial. Deaths were recorded when the flies were transferred.

### Statistics and data presentation

All figures and data were generated and analyzed using GraphPad Prism (v9.1.2). For experiments with two groups, a Student's t-test was performed. For experiments with three or more groups, a one-way ANOVA with Tukey HSD post hoc test was performed. For fat breakdown experiments, a two-way ANOVA was used to determine the interaction between genotype and time. Starvation resistance and lifespan statistics were performed using RStudio and a script for a log-rank test with Bonferroni's correction for multiple comparisons. Note, the lowest p-value given by RStudio is $2.0×10^{-16}$. The below packages and script were used: library ("survminer") library ("survival") data <- read.csv("xxx. csv") survfit(Surv(time, event)~ genotype, data) pairwise_survdiff(Surv(time, event)~ genotype, data, p.adjust.method = "bonferroni") summary (data).

## Acknowledgements

The authors would like to thank Dr. Ronald Kühnlein for *Akh*[A], *AkhR*[rev], *AkhR*[1], *bmm*[1], and *AkhR*[1];*bmm*[1] fly strains. The authors would also like to thank Dr. Irene Miguel-Aliaga for the *tra*[KO] and *tra*[F K-IN] strains, and Dr. Claire Thomas for *Mex-GAL4.* Stocks obtained from the Bloomington *Drosophila* Stock Center (NIH P40OD018537) were used in this study. The authors thank the TRiP at Harvard Medical School (NIH/NIGMS R01-GM084947) for providing transgenic RNAi fly stocks and/or plasmid vectors used in this study. FlyBase is supported by a grant from the National Human Genome Research Institute at the U.S. National Institutes of Health (U41 HG000739) and by the British Medical Research Council (MR/N030117/1). Funding for this study was provided by grants to EJR from the Canadian Institutes for Health Research (CIHR, PJT-153072), the CIHR Sex and Gender Science Chair Program (GS4-171365), the Natural Sciences and Engineering Research Council of Canada (NSERC, RGPIN-2016-04249), the Michael Smith Foundation for Health Research (16876), and the Canadian Foundation for Innovation (JELF-34879). LWW was supported by a British Columbia Graduate Scholarship Award and a 1-year CELL Fellowship from UBC. JWM and PB were each supported by a 4-year CELL Fellowship from UBC. The authors acknowledge that our research takes place on the traditional, ancestral, and unceded territory of the Musqueam people; a privilege for which we are grateful.

# Additional information

## Funding

| Funder | Grant reference number | Author |
| --- | --- | --- |
| Canadian Institutes of Health Research | PJT-153072 | Elizabeth J Rideout |
| CIHR Sex and GenderScience chair program | GS4-171365 | Elizabeth J Rideout |
| Natural Sciences and Engineering Research Council of Canada | RGPIN-2016-04249 | Elizabeth J Rideout |
| Michael Smith Foundation for Health Research | 16876 | Elizabeth J Rideout |
| Canadian Foundation for Innovation | JELF-34879 | Elizabeth J Rideout |

The funders had no role in study design, data collection and interpretation, or the decision to submit the work for publication.

## Author contributions

Lianna W Wat, Conceptualization, Formal analysis, Investigation, Visualization, Writing – original draft, Writing – review and editing; Zahid S Chowdhury, Conceptualization, Formal analysis, Investigation; Jason W Millington, Puja Biswas, Investigation; Elizabeth J Rideout, Conceptualization, Funding acquisition, Project administration, Supervision, Validation, Writing – original draft, Writing – review and editing

## Author ORCIDs

Lianna W Wat (iD) http://orcid.org/0000-0001-6998-0594
Jason W Millington (iD) http://orcid.org/0000-0003-4330-2431
Elizabeth J Rideout (iD) http://orcid.org/0000-0003-0012-2828

## Decision letter and Author response

Decision letter https://doi.org/10.7554/eLife.72350.sa1
Author response https://doi.org/10.7554/eLife.72350.sa2

# Additional files

## Supplementary files

- Supplementary file 1. 1 p-values.
- Supplementary file 2. Raw data.
- Supplementary file 3. Primers.
- Supplementary file 4. Fly food media.
- Transparent reporting form

## Data availability

Details of all statistical tests and p-values are in Supplementary file 1. All raw data generated in this study are in Supplementary file 2. All primer sequences are in Supplementary file 3. Fly food media recipe is in Supplementary file 4. Original image files for all images in this study are in their respective Source Data files.

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
