## [Decision Letter]

**Acceptance summary:**

Females of many animal species are known to store more fat, which is necessary for sustaining reproduction. In this study, the authors show that the gene transformer controls differences in fat deposition between males and females by regulating the production of the highly conserved adipokinetic hormone, a hormone involved in regulating fat storage. The experiments are very well-conceived and well-executed and data justify major conclusions. This work will be of interest to those studying sexual dimorphism, metabolism, and the genetic regulation of life-history traits.

**Decision letter after peer review:**

Thank you for submitting your article "Sex determination gene *transformer* regulates the male-female difference in *Drosophila* fat storage via the adipokinetic hormone pathway" for consideration by *eLife*. Your article has been reviewed by 3 peer reviewers, one of whom is a member of our Board of Reviewing Editors, and the evaluation has been overseen by a Senior Editor.

*Reviewer #1:*

The sex-determination role of tra or the metabolic function of adipokinetic hormone (AKH) has been extensively studied in previous studies; however, the genetic link between tra and AKH in the metabolic dimorphism has not been shown. This study identified genetic interactions between tra and AKH that differentially decide the amount of fat storage in males or females. Experiments are well-designed, performed, and analyzed, and the manuscript is clearly written.

In this study, Rideout and colleagues investigated a role for transformer (tra) in the female-biased fat storage and identified that a functional Tra promotes fat storage via inhibition of Akh producing cells in females. This mechanism is valid only in females and thus males activate Akh to mobilize triglycerides. This study unravels a novel mechanism underlying the male-female differences in fat storage and provides valuable insights into the sexual dimorphism in fat metabolism.

1. Authors showed that expression levels of Akh and AkhR are upregulated in males (Figure 3A-B) and males exhibit higher activities of APC (Figure 3E-H'). However, these two readouts could be distinctive and need to be described in more detail. For example, well-known regulators of Akh release are energy sensors, K^+^ ATP channels, or AMPK (Kim and Rulifson, 2004; Braco et al., 2012), and Tra could impact the expression of sensors or the ratio of cellular AMP/ATP in addition to the Akh/AkhR control.

2. Related to the above concern, it is not clear which tissue does Akh target to cause the sex-specific bias. One can simply guess it could be the fat body where Akh acts on but other tissues including gonads could also be a target of Akh given the changes in fertility and fecundity. This needs to be further strengthened.

3. Tra may function in both directions of preventing excessive fat breakdown and increasing storage, both of which cases lead to the single point reading of storage phenotype. This creates a little confusion over causation, which the authors need to explain or discuss.

4. A good schematic diagram will give a pictorial overview of key conclusions.

*Reviewer #2:*

Males and female animals differ in a number of key life history traits, including body size, lifespan, and metabolism. The study by Wat and co-authors aims to uncover the pathways that generate differences in fat storage between male and female *D. melanogaster* flies. In this manuscript, the authors show that a key gene in the sex determination pathway, transformer, which is normally expressed only in females, is responsible for higher fat storage in females. They find that transformer regulates fat storage by reducing the production of adipokinetic hormone (AKH). Either inducing transformer expression or reducing AKH expression in males leads to increased fat deposition.

The experiments are very well conceived and well executed. I find the claims to be well supported by the results. I feel that this work will be of interest to those studying sexual dimorphism, metabolism, and the genetic regulation of life history traits.

This is an incredibly thorough, well-conceived, and well-executed study uncovering the pathways that generate differences in fat storage between male and female *D. melanogaster* flies. In this manuscript, the authors show that transformer, normally expressed in females, is responsible for regulating differences in adipokinetic hormone (AKH) production. Males have higher AKH production, which leads to reduced fat storage. The experiments are very well controlled and I find the results to be incredibly sound. I have a few comments that I think might help, but really have more to do with clarifying the results and methods. In my opinion, the study is of excellent quality and does not require any additional experiments.

I'm not sure I understand the difference between Figure 1A and 1B. Figure 1A compares tra1/Df(3L)st-j7 virgin females to *w1118* virgin females. Figure 1B talks about the same tra1/Df(3L)st-j7 and *w1118* genotypes, but in this case without gonads/ovaries. In the figure legend it states that the ovaries were excised for tra1/Df(3L)st-j7. I assume that this was also done on the *w1118* females, but a bit more description as to how this experiment was done would be helpful here. I think it needs to be explicitly covered in the methods section as well.

Also, I find the comparisons between fat breakdown in starved males and females in Lines 223-226 a bit hard to follow. I think explaining the experiment in a bit more detail would help. Part of what I think I didn't understand was how you differentiated between whole body triglyceride quantifications, termed fat storage, and quantifications of fat breakdown. I think the difference relates to the quantifications of body fat in well fed versus starved animals, but this isn't explicitly stated nor are the starvation protocols outlined (they also aren't described in detail in the methods).

Figure Suppl 1D: It's true that ecdysone regulated genes are not upregulated in the traFK-IN males, but all genes sampled show significant downregulation. This isn't discussed in the results text. Males also produce ecdysone, albeit at lower concentrations, which is required for germline stem cell maintenance in the testes. Why do you think that traFK-IN reduces ecdysone signalling?

The Akh GAL4> UAS CaLexA experiment is extremely cool! I especially appreciate the care you took to make sure that the GAL4 was expressed at equivalent levels across sexes.

*Reviewer #3:*

Wat et al., investigated how sex-dependent differences in the lipid storage amounts of fruit flies arise. They find that the sex determination gene transformer (tra) is playing a key role in this process and reveal that sex-dependent tra activity in the neuroendocrine cells which secrete the adipokinetic hormone (Akh) are mediating this function. By various genetic manipulations targeting tra or Akh activity they dissect the impact of tra and Akh on the sex-biased lipid storage amounts. Altogether, the study advances our understanding of sex-biased metabolic regulation. As the Akh signaling pathway shares similarities with the mammalian β-adrenergic and glucagon systems, the findings of the present study might also present indications for similar regulatory mechanisms in vertebrates.

The conclusions reached on the basis of the experimental data are justified and the authors use a rich set of methods and genetics systems to answer their questions. Still, some aspects should be explained and discussed in greater detail.

– The magnitude of effects by seemingly comparable methods varies more than anticipated. An example are the data shown in Figures 1D and E as well as Figure S1E and F. In these panels, the authors compare a knock-in of tra to the ubiquitous Gal4/UAS mediated (and presumably strong) overexpression of tra. The knock-in was used "because high levels of Tra overexpression may influence viability" (line 240). While the authors do not mention or determined any such effects, as for example a decreased viability, the male fat storage increase following the overexpression is much weaker as compared to the knock-in situation, which should result in physiological expression levels. The authors should discuss possible explanations for this discrepancy and e.g. compare expression levels of the different constructs in key tissues or whole animal extracts.

– In their Gal4/UAS based screen for the phenocritic tissue of tra activity, the authors focus on the adipokinetic hormone producing (APC) cells. Yet, browsing the various results shown in the supplements for Figure 2, different additional sites of expression also affect starvation resistance. While some of these affect starvation resistance in the opposite direction of the neuronal / APC cell expression (e.g. in the mushroom body), others show the same phenotypic directionality (e.g. expression in the glia) and are similarly male-biased. Thus, with this data the focusing on the APC cells is not totally clear to me and the most likely more complex regulation should at least be discussed further.

– Sometimes the use of varying experimental systems is not obvious and the authors should help readers to better understand why the one or the other system was used. A prominent example is in the end, when the authors tested for an impact on life history traits of Akh alterations. Here, they used for the fertility quantifications a mutation of the Akh and for the lifespan the Akh-Gal4 mediated expression of the Kir2.1 potassium channel. How do the fertility measurements look following overexpression of the channel or what is the lifespan of flies lacking the adipokinetic hormone?

– In their study, the authors exclusively focus on the lipid metabolism. Yet, Akh is also long-known for its role in regulating carbohydrate metabolism. The authors should consider including this aspect at least in the discussion.

– In line 107 the authors write "mammalian Lsd-1". Use of the mammalian Perilipin1 name, however, is advised.

– In Figure S1D all genes are downregulated in the tra knock-in flies. The text only states that "no ecdysone target genes were upregulated" (line 250). The authors should be more precise here.

---

## [Author Response]

Reviewer #1:The sex-determination role of tra or the metabolic function of adipokinetic hormone (AKH) has been extensively studied in previous studies; however, the genetic link between tra and AKH in the metabolic dimorphism has not been shown. This study identified genetic interactions between tra and AKH that differentially decide the amount of fat storage in males or females. Experiments are well-designed, performed, and analyzed, and the manuscript is clearly written.

We thank the Reviewer for their positive comments on our paper, and for their suggestions on ways to improve the manuscript.

In this study, Rideout and colleagues investigated a role for transformer (tra) in the female-biased fat storage and identified that a functional Tra promotes fat storage via inhibition of Akh producing cells in females. This mechanism is valid only in females and thus males activate Akh to mobilize triglycerides. This study unravels a novel mechanism underlying the male-female differences in fat storage and provides valuable insights into the sexual dimorphism in fat metabolism.1. Authors showed that expression levels of Akh and AkhR are upregulated in males (Figure 3A-B) and males exhibit higher activities of APC (Figure 3E-H'). However, these two readouts could be distinctive and need to be described in more detail. For example, well-known regulators of Akh release are energy sensors, K^+^ ATP channels, or AMPK (Kim and Rulifson, 2004; Braco et al., 2012), and Tra could impact the expression of sensors or the ratio of cellular AMP/ATP in addition to the Akh/AkhR control.

We thank the Reviewer for pointing out we did not clearly state that higher *Akh/AkhR* mRNA levels and APC activity are not necessarily linked. We added the following text to the revised manuscript to clarify this point:

“Indeed, while our identification of a role for APC sexual identity in regulating the male-female difference in fat storage represents a significant step forward in understanding how sex differences in neurons influence metabolic traits, more knowledge is needed of how Tra regulates sexual dimorphism in this critical neuronal subset. For example, while we show that females normally have lower *Akh* mRNA levels and APC activity, it is unclear how the presence of Tra regulates these distinct traits. Tra may regulate *Akh* mRNA levels via known target genes *fruitless* (*fru*; FBgn0004652) and *doublesex* (*dsx*; FBgn0000504) (Heinrichs et al., 1998; Ryner et al., 1996; Hoshijima et al., 1991; Inoue et al., 1992; Burtis and Baker, 1989; Nagoshi et al., 1998*)*, or alternatively through a *fru*- and *dsx*-independent pathway (Rideout et al., 2015; Hudry et al., 2016). To influence the sex difference in APC activity and Akh release, Tra may regulate factors such as ATP-sensitive potassium (KATP) channels and 5’ adenosine monophosphate-activated protein kinase (AMPK)-dependent signaling, both of which are known to modulate APC activity (Kim and Rulifson, 2004; Braco et al., 2012). Future studies will therefore need to investigate Tra-dependent changes to KATP channel expression and function in APCs, and characterize Tra’s effects on ATP levels and AMPK signaling within APCs.”

2. Related to the above concern, it is not clear which tissue does Akh target to cause the sex-specific bias. One can simply guess it could be the fat body where Akh acts on but other tissues including gonads could also be a target of Akh given the changes in fertility and fecundity. This needs to be further strengthened.

To clarify the tissue(s) potentially targeted by Akh to promote fat breakdown, we added references that describe the fat body expression of *AkhR* (Grönke et al., 2007; Bharucha et al., 2008), and which demonstrate that fat body *AkhR* expression largely rescues the excess triglyceride storage observed in flies carrying *AkhR* loss-of-function mutations (Bharucha et al., 2008):

“Another important point to address in future studies will be confirming results from previous studies that the fat body is the main anatomical focus of Akh-dependent regulation of fat storage (Grönke et al., 2007; Bharucha et al., 2008).”

We also added text to acknowledge the important possibility that Akh binds to the gonads to influence fertility and fecundity:

“Given that we and others show Akh affects fertility and fecundity (Liao et al., 2021), future studies will need to determine whether these phenotypes are due to Akh-dependent regulation of fat metabolism, or due to direct effects of Akh on gonads.”

3. Tra may function in both directions of preventing excessive fat breakdown and increasing storage, both of which cases lead to the single point reading of storage phenotype. This creates a little confusion over causation, which the authors need to explain or discuss.

To address this key point, we added the following text to the revised manuscript:

“This line of enquiry will also clarify the underlying processes that support increased fat storage in females. At present, it remains unclear whether the higher whole-body fat storage in females is caused by lower fat breakdown (Wat et al., 2020), increased lipogenesis, or both. Given that Akh pathway activity plays a role in regulating both lipolysis and lipogenesis in *Drosophila* and other insects (Grönke et al., 2007; Lee and Goldsworthy, 1995; Lorenz, 2001), it will be important to identify the cellular mechanism by which Akh contributes to the sex difference in fat storage.”

4. A good schematic diagram will give a pictorial overview of key conclusions.

We added an additional figure to the revised version of the manuscript to provide a schematic representation of our model on how the Tra-Akh axis regulates the sex difference in fat storage (Figure 7).

Reviewer #2:Males and female animals differ in a number of key life history traits, including body size, lifespan, and metabolism. The study by Wat and co-authors aims to uncover the pathways that generate differences in fat storage between male and female *D. melanogaster* flies. In this manuscript, the authors show that a key gene in the sex determination pathway, transformer, which is normally expressed only in females, is responsible for higher fat storage in females. They find that transformer regulates fat storage by reducing the production of adipokinetic hormone (AKH). Either inducing transformer expression or reducing AKH expression in males leads to increased fat deposition.The experiments are very well conceived and well executed. I find the claims to be well supported by the results. I feel that this work will be of interest to those studying sexual dimorphism, metabolism, and the genetic regulation of life history traits.This is an incredibly thorough, well-conceived, and well-executed study uncovering the pathways that generate differences in fat storage between male and female *D. melanogaster* flies. In this manuscript, the authors show that transformer, normally expressed in females, is responsible for regulating differences in adipokinetic hormone (AKH) production. Males have higher AKH production, which leads to reduced fat storage. The experiments are very well controlled and I find the results to be incredibly sound. I have a few comments that I think might help, but really have more to do with clarifying the results and methods. In my opinion, the study is of excellent quality and does not require any additional experiments.

We thank the Reviewer for their positive comments on the manuscript.

I'm not sure I understand the difference between Figure 1A and 1B. Figure 1A compares tra1/Df(3L)st-j7 virgin females to w1118 virgin females. Figure 1B talks about the same tra1/Df(3L)st-j7 and w1118 genotypes, but in this case without gonads/ovaries. In the figure legend it states that the ovaries were excised for tra1/Df(3L)st-j7. I assume that this was also done on the w1118 females, but a bit more description as to how this experiment was done would be helpful here. I think it needs to be explicitly covered in the methods section as well.

In the revised version of the manuscript, we now explicitly state that Figure 1B compares whole-body triglyceride levels between two genotypes of females (*tra* mutants and controls) where both genotypes have had their gonads removed. This allowed us to compare how much fat was present in non-gonadal tissues between genotypes.

“While previous studies show the ovaries store a small amount of triglyceride (Sieber and Spradling, 2015; Wat et al., 2020), Tra’s effect on whole-body triglyceride storage was not explained by the absence of ovaries in females lacking Tra function: whole-body fat storage was significantly reduced in *tra^1^/Df(3L)st-j7* mutant females with excised gonads compared with *w1118* control females with excised ovaries (Figure 1B).”

We also added text describing the protocol in the methods section:

“Gonad excision. 5-day old adult flies were individually anesthetized with CO_2_. The gonads or ovaries were removed from the distal end of the abdomen in cold 1X PBS and the carcass was snap-frozen in 1.5 ml microcentrifuge tubes on dry ice.”

Also, I find the comparisons between fat breakdown in starved males and females in Lines 223-226 a bit hard to follow. I think explaining the experiment in a bit more detail would help. Part of what I think I didn't understand was how you differentiated between whole body triglyceride quantifications, termed fat storage, and quantifications of fat breakdown. I think the difference relates to the quantifications of body fat in well fed versus starved animals, but this isn't explicitly stated nor are the starvation protocols outlined (they also aren't described in detail in the methods).

In the revised version of the manuscript, we added text in the Results section to clarify how we represent fat breakdown:

“Next we asked whether Tra function also contributes to reduced fat breakdown post-starvation in females compared with males. To quantify fat breakdown, we measured whole-body triglyceride levels at 0 hr and 24 hr post-starvation, and calculated the percent change in whole-body triglyceride levels between time points.”

In the methods section we added the following text:

“To measure fat breakdown post-starvation, biological replicates consisting of 5 flies each were collected in 1.5 ml microcentrifuge tubes and snap frozen on dry ice at 0 hr and 24 hr post-starvation. The percent change in fat storage between time points was calculated to determine fat breakdown over time.”

Figure Suppl 1D: It's true that ecdysone regulated genes are not upregulated in the traFK-IN males, but all genes sampled show significant downregulation. This isn't discussed in the results text. Males also produce ecdysone, albeit at lower concentrations, which is required for germline stem cell maintenance in the testes. Why do you think that traFK-IN reduces ecdysone signalling?

We were also interested in why ecdysone target genes were downregulated in *tra^F^*
^K-IN^ males. We added text in the Results section to explicitly state this result for readers, and to suggest further work will be needed to understand the underlying mechanism:

“The elevated fat storage in *tra^F K-IN^* males also cannot be attributed to ecdysone production by the rudimentary ovaries, as no ecdysone target genes were upregulated (Figure 1 —figure supplement 1D) (Sieber and Spradling, 2015); however, future studies will need to address why these *tra^F K-IN^* males showed significant ecdysone target gene downregulation.”

Reviewer #3:Wat et al., investigated how sex-dependent differences in the lipid storage amounts of fruit flies arise. They find that the sex determination gene transformer (tra) is playing a key role in this process and reveal that sex-dependent tra activity in the neuroendocrine cells which secrete the adipokinetic hormone (Akh) are mediating this function. By various genetic manipulations targeting tra or Akh activity they dissect the impact of tra and Akh on the sex-biased lipid storage amounts. Altogether, the study advances our understanding of sex-biased metabolic regulation. As the Akh signaling pathway shares similarities with the mammalian β-adrenergic and glucagon systems, the findings of the present study might also present indications for similar regulatory mechanisms in vertebrates.The conclusions reached on the basis of the experimental data are justified and the authors use a rich set of methods and genetics systems to answer their questions. Still, some aspects should be explained and discussed in greater detail.

We thank the Reviewer for their positive comments on our manuscript, and for their thoughtful suggestions.

– The magnitude of effects by seemingly comparable methods varies more than anticipated. An example are the data shown in Figures 1D and E as well as Figure S1E and F. In these panels, the authors compare a knock-in of tra to the ubiquitous Gal4/UAS mediated (and presumably strong) overexpression of tra. The knock-in was used "because high levels of Tra overexpression may influence viability" (line 240). While the authors do not mention or determined any such effects, as for example a decreased viability, the male fat storage increase following the overexpression is much weaker as compared to the knock-in situation, which should result in physiological expression levels. The authors should discuss possible explanations for this discrepancy and e.g. compare expression levels of the different constructs in key tissues or whole animal extracts.

We thank the Reviewer for pointing out the discrepancy in the magnitude of effects on fat storage mediated by global Tra expression in males compared with males carrying the *tra^F K-IN^* allele. We added text to the revised manuscript to ensure readers are aware of this interesting point:

“While these data indicate that gain of a functional Tra protein in males promotes whole-body fat storage, we note that the magnitude of the increase in fat storage was higher in *tra^F K-IN^* males. The reason for this discrepancy between genotypes is not clear, therefore, future studies will need to compare *tra* expression levels and tissue distribution between *da-GAL4>UAS-tra^F^* males and *tra^F K-IN^* males.”

– In their Gal4/UAS based screen for the phenocritic tissue of tra activity, the authors focus on the adipokinetic hormone producing (APC) cells. Yet, browsing the various results shown in the supplements for Figure 2, different additional sites of expression also affect starvation resistance. While some of these affect starvation resistance in the opposite direction of the neuronal / APC cell expression (e.g. in the mushroom body), others show the same phenotypic directionality (e.g. expression in the glia) and are similarly male-biased. Thus, with this data the focusing on the APC cells is not totally clear to me and the most likely more complex regulation should at least be discussed further.

We thank the Reviewer for suggesting we further clarify the rationale for our focus on the APCs. To address this point we added the following text to the revised manuscript:

“Although we note that Tra expression in additional neurons and in glia also affected starvation resistance (Figure 2 —figure supplement 2D; Figure 2 —figure supplement 4D), suggesting the regulation of fat metabolism by Tra function in neurons is complex, the central role of the APCs in regulating fat metabolism prompted a more detailed investigation into Tra’s function in these neurons.”

– Sometimes the use of varying experimental systems is not obvious and the authors should help readers to better understand why the one or the other system was used. A prominent example is in the end, when the authors tested for an impact on life history traits of Akh alterations. Here, they used for the fertility quantifications a mutation of the Akh and for the lifespan the Akh-Gal4 mediated expression of the Kir2.1 potassium channel. How do the fertility measurements look following overexpression of the channel or what is the lifespan of flies lacking the adipokinetic hormone?

We thank the Reviewer for this comment. In the revised text we clarify the effects of each genetic system on Akh signaling (e.g. loss-of-function mutation, APC ablation, APC silencing) when the experimental system is first used.

We also added text to the revised manuscript to suggest future studies should be done to determine the effects of multiple Akh pathway manipulations on fertility, lifespan, and fecundity.

“This suggests that while low Akh activity in females promotes fertility, this benefit comes at the cost of a shorter lifespan, a possibility that will be explored in future studies using additional strains to genetically augment, or inhibit, Akh pathway activity (e.g. APC activation, Akh mutants).”

– In their study, the authors exclusively focus on the lipid metabolism. Yet, Akh is also long-known for its role in regulating carbohydrate metabolism. The authors should consider including this aspect at least in the discussion.

We thank the Reviewer for pointing out the important role of Akh in regulating carbohydrate metabolism. To correct this omission, we added the following text to the revised version:

“Similarly, while Akh has been linked with the regulation of lifespan (Bednářová et al., 2018; Liao et al., 2021), carbohydrate metabolism (Lee and Park, 2004; Kim and Rulifson, 2004),….”

And also “Additional work is therefore needed to determine how changes to Akh pathway function affect physiology, carbohydrate levels, development*,*…”

– In line 107 the authors write "mammalian Lsd-1". Use of the mammalian Perilipin1 name, however, is advised.

We corrected this error.

– In Figure S1D all genes are downregulated in the tra knock-in flies. The text only states that "no ecdysone target genes were upregulated" (line 250). The authors should be more precise here.

We were also interested in why ecdysone target genes were downregulated in *tra^F^*
^K-IN^ males. We added text in the Results section to explicitly state this result for readers, and to suggest further work will be needed to understand the underlying mechanism:

“The elevated fat storage in *tra^F K-IN^* males also cannot be attributed to ecdysone production by the rudimentary ovaries, as no ecdysone target genes were upregulated (Figure 1 —figure supplement 1D) (Sieber and Spradling, 2015); however, future studies will need to address why these *tra^F K-IN^* males show significant ecdysone target gene downregulation.”